# ECO: Evolving Core Knowledge for Efficient Transfer

**Fu Feng**[1,2]   **Yucheng Xie**[1,2]   **Ruixiao Shi**[1,2]   **Jianlu Shen**[1,2]   **Jing Wang**[1,2*]   **Xin Geng**[1,2*]

[1]School of Computer Science and Engineering, Southeast University, Nanjing, China
[2]Key Laboratory of New Generation Artificial Intelligence Technology and Its Interdisciplinary
Applications (Southeast University), Ministry of Education, China
{fufeng, xieyc, eric_xiao, jlshen, wangjing91, xgeng}@seu.edu.cn

## Abstract

Knowledge in modern neural networks is often entangled and structurally opaque, making current transfer methods—typically based on reusing entire parameter sets—inefficient and inflexible. Efforts to improve flexibility by reusing partial parameters frequently depend on handcrafted heuristics or rigid structural assumptions, which constrain generalization. In contrast, biological evolution enables efficient knowledge transfer by encoding only essential information into genes through iterative refinement under environmental pressure. Inspired by this principle, we propose **ECO**, a framework that **E**volves **CO**re knowledge into modular, reusable neural components—termed *learngenes*—through similar evolutionary dynamics. To this end, we redefine learngenes as neural circuits and introduce Genetic Transfer Learning (GTL), a biologically inspired paradigm that establishes a genetic mechanism within neural networks in the context of supervised learning. GTL simulates evolutionary processes by generating diverse network populations, selecting high-performing individuals, and transferring their learngenes to subsequent generations. Through iterative refinement, GTL enables learngenes to accumulate transferable common knowledge. Extensive experiments show that ECO achieves efficient initialization and strong generalization across diverse models and tasks, while significantly reducing computational and memory costs compared to conventional methods.

## 1   Introduction

Despite the remarkable progress of modern neural networks across a wide range of tasks [1, 61, 13], the internal organization of knowledge within these models remains largely opaque and poorly structured. As a result, conventional transfer learning approaches—such as full fine-tuning or parameter-efficient techniques like LoRA [22, 17]—primarily rely on reusing entire parameter sets from large-scale pre-trained models [6, 35]. Although effective in reducing task-specific training overhead [5, 87], these methods implicitly assume that knowledge is globally entangled and uniformly distributed, with little consideration for its internal structure or modular organization. Consequently, they often suffer from limited adaptability across models of varying sizes and architectures [12, 79, 77], and are highly susceptible to domain shifts, frequently resulting in biased [49, 31] or even detrimental transfer [50, 73, 32], as shown in Figure 1a.

Biological systems, in contrast, offer a compelling model for efficient and generalizable knowledge transfer. Rather than replicating entire structural configurations, they encode essential knowledge into compact and inheritable units known as genes, which capture core functionality and support adaptation across generations and environments. These genes act as reusable blueprints [82, 3, 65], guiding the development of neural circuits that are both robust and efficient, as illustrated in Figure 1b.

---

[*]Corresponding authors

39th Conference on Neural Information Processing Systems (NeurIPS 2025).

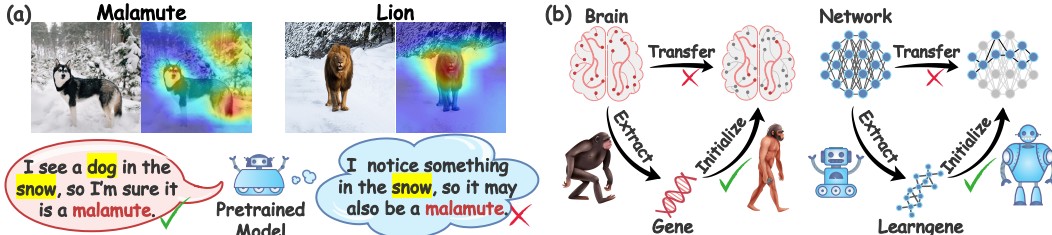

Figure 1: (a) Transferring excessive knowledge may introduce bias and lead to negative transfer. (b) In contrast, biological genes transfer only survival-essential knowledge to initialize certain innate neural circuits. Inspired by this, we propose encoding core transferable knowledge into modular neural circuits (termed *learngenes*, see Figure 2) to enable efficient knowledge transfer.

Motivated by this principle, recent work has explored biologically inspired approaches to knowledge transfer, most notably through the *Learngene* framework [11, 70]. Unlike conventional parameter reuse, learngenes aim to support structured and composable transfer by encapsulating common knowledge into modular neural fragments, enabling flexible initialization across tasks and models of diverse scales. Early methods such as Heur-LG [68] and Auto-LG [70] identify transferable components using gradient-based heuristics or meta-learning strategies. More recent approaches, including TLEG [75] and WAVE [12], incorporate structural priors—such as linear constraints or Kronecker decompositions—to enhance the modular organization of learned representations. Despite their improved flexibility and efficiency, they often depend on handcrafted heuristics or rigid structural assumptions, limiting their adaptability and generalization across diverse tasks and architectures.

In contrast, essential knowledge encoded in biological genes is not organized by predefined rules, but emerges through iterative refinement under environmental pressures such as mutation, selection, and inheritance. This perspective motivates a shift toward understanding how transferable knowledge can be modularly organized within neural networks through similar evolutionary dynamics, rather than imposed through handcrafted heuristics or rigid structural assumptions (e.g., linear constraints). To this end, we propose **ECO**, a biologically inspired framework that explores whether neural networks can autonomously condense transferable core knowledge and encapsulate it into compact, reusable modules through population-based adaptation and feedback-driven refinement.

At the core of ECO is a redefinition of the learngene structure as continuous neural circuits, such as interconnected kernel sets in CNNs, which mirror the structure of innate pathways shaped by genetic encoding in biological systems (see Figure 1b). To automatically identify and encapsulate highly transferable core knowledge within neural circuits, we extend the evolutionary paradigm introduced in GRL [11] and propose **Genetic Transfer Learning** (GTL), an evolutionary framework that simulates genetic inheritance in neural networks in the context of supervised learning. GTL maintains a population of models, each trained on randomly sampled tasks. High-performing individuals are selected via tournament evaluation, and their learngenes are inherited and refined across generations, with mutation introduced to maintain diversity and promote exploration. This Lamarckian process progressively condenses core knowledge into learngenes, enabling efficient, scalable, and generalizable transfer across models and tasks.

We evolve learngenes for 250 generations in VGG11 and ResNet12 on CIFAR-FS and *mini*ImageNet, and for 100 generations in ResNet50 and MobileNetV3-Large on ImageNet. Extensive experiments demonstrate that ECO consistently outperforms existing methods across a wide range of downstream architectures and tasks. Compared to full fine-tuning, ECO achieves a $14.5\times$ reduction in storage, highlighting its efficiency in knowledge transfer. Additionally, ECO supports $O(1)$ initialization across models of varying scales, enabling fast deployment without additional costs. Notably, ECO exhibits strong data efficiency, maintaining robust performance even in low-resource regimes.

Our main contributions are as follows: 1) We propose ECO, a novel knowledge transfer method that adaptively identifies core transferable knowledge across architectures and tasks, without relying on handcrafted heuristics or structural constraints. 2) We introduce Genetic Transfer Learning (GTL), a biologically inspired evolutionary paradigm for supervised learning that encapsulates core transferable knowledge into modular neural circuits (i.e., learngenes) through large-scale population-based training and inheritance. 3) We validate the scalability and generality of ECO across diverse tasks, model sizes, and architectures, achieving state-of-the-art performance in both accuracy and resource efficiency.

## 2  Related Work

**Efficient Knowledge Transfer**    Transfer learning enhances performance on target domains by leveraging knowledge from source domains [86, 23]. Traditional methods, such as fine-tuning [18, 87], are constrained by the fixed architecture and size of pre-trained models. Knowledge distillation [57, 67, 16] offers structural flexibility but remains computationally intensive. Recent methods aim to enhance efficiency by transferring compact and reusable knowledge representations (see Appendix A.1). Learngene-based approaches, including Heur-LG [68] and Auto-LG [70], identify transferable components via heuristic strategies. Others, such as TLEG [75], WAVE [12], and related studies [37, 69, 78], integrate pre-trained knowledge via structural priors like low-rank decompositions. Alternative paradigms explore hypernetwork-based parameter generation [29, 30] or rule-based parameter reuse [80, 33, 83]. In contrast, ECO extends the population-based learning paradigm of GRL [11] to CNNs in the context of supervised learning, adaptively evolving core transferable knowledge through mutation-driven and survival-based selection. This process mitigates manual bias and enhances generalization across architectures and domains.

**Evolutionary Learning**    Evolutionary Learning (EL) solves optimization problems through stochastic search inspired by biological evolution [60]. Related algorithms [56, 55, 43, 53, 44] typically encode candidate solutions—such as network parameters or architectures—into gene-like representations (e.g., binary strings) [40, 63, 45, 9, 84], evolving them to maximize task-specific performance. While inspired by evolutionary principles, ECO diverges from traditional evolutionary learning by prioritizing knowledge inheritance over solution optimization. Notably, instead of searching for task-specific solutions, ECO establishes a genetic transfer mechanism that accumulates core knowledge transferable across diverse models and tasks. In this paradigm, learngenes function not as solution encodings, but as compact carriers of core knowledge condensed from diverse tasks. This shift enables scalable generalization without reliance on problem-specific heuristics.

## 3  Methods

We reformulate CNNs in terms of kernel units and define the core operations of learngenes, including their representation, mutation, and inheritance. We then introduce the Genetic Transfer Learning (GTL) for extracting learngenes through evolutionary processes.

### 3.1  Preliminary

Consider a CNN $\mathcal{N}$ with $N_L$ convolutional layers. The $l$-th layer $L_l \in \mathbb{R}^{N_F^{(l)} \times N_K^{(l)} \times \kappa \times \kappa}$ comprises $N_F^{(l)}$ filters $F_{l,f}$, where each filter contains $N_K^{(l)}$ kernels $K_{l,f,k} \in \mathbb{R}^{\kappa \times \kappa}$. These kernels capture spatial features at various levels of abstraction, enabling hierarchical representation learning. Accordingly, the trainable parameters of the entire network can be represented as a unified set of kernels:

$$\mathcal{N} = \{K_{l,f,k} | l \in [1, N_L], f \in [1, N_F^{(l)}], k \in [1, N_K^{(l)}]\} \tag{1}$$

Given an input feature map $I_l \in \mathbb{R}^{D \times H \times W}$ to layer $L_l$, the convolution produces an output $I_{l+1} \in \mathbb{R}^{N_F^{(l)} \times H \times W}$, which serves as input to layer $L_{l+1}$. To ensure valid feature propagation, CNNs enforce channel-wise consistency by matching the number of kernels in $L_{l+1}$ with the number of filters in $L_l$:

$$N_K^{(l+1)} = N_F^{(l)} \tag{2}$$

### 3.2  Basic Operations for Learngenes

#### 3.2.1  Form of Learngenes in CNNs

In biological neural systems, innate neural circuits are established at birth under the guidance of genes, providing newborns with strong inherent learning abilities [74, 42, 82]. Motivated by this biological foundation, we interpret neural circuits in convolutional neural networks (CNNs) as structured subnetworks [59], each composed of a set of interconnected kernels that collectively implement a continuous input-output transformation. These subnetworks serve as functional units responsible for localized computation and information flow within the network.

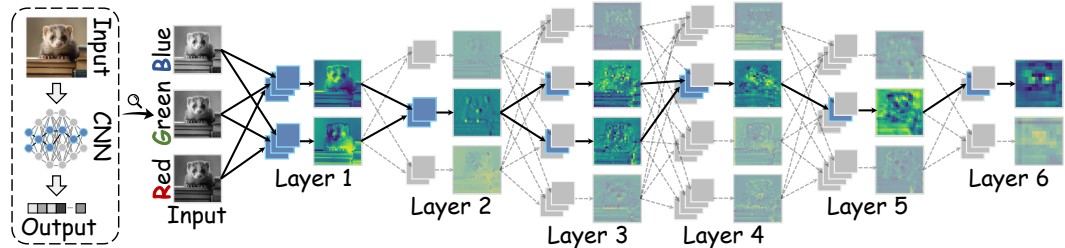

Figure 2: Learngenes in ECO are abstracted as complete neural circuits composed of selected kernels within filters. ■ ∈ $\mathcal{G}$ represents the learngene kernel, while ▧ is normal kernel that is random initialized. ➜ denotes the continuous feature mapping path extracted from learngenes.

Building upon this abstraction, we define learngenes as modular neural circuits within CNNs that encapsulate transferable and reusable knowledge. Formally, a learngene is defined as:

$$\mathcal{G} = \{K_{l,f,k} | l \in [1, N_L], f \in \mathcal{F}_l, k \in \mathcal{K}_l\} \tag{3}$$

where $\mathcal{F}_l$ and $\mathcal{K}_l$ denote the selected indices of filters and kernels in the $l$-th layer, respectively.

To preserve the interconnection among kernels and ensure uninterrupted continuity in feature transformation, learngenes follow the structural alignment principle inherent to CNNs (Eq. (2)). Specifically, we enforce the constraint:

$$\mathcal{K}_{l+1} = \mathcal{F}_l \tag{4}$$

This condition preserves channel-wise consistency, enabling layer-wise modification through $\mathcal{F}_l$ while $\mathcal{K}_l$ is automatically inferred via Eq. (4), ensuring coherent inter-layer connectivity of learngenes.

### 3.2.2 Mutation of Learngenes

In biological systems, evolution is driven by structural mutations that progressively refine genetic traits [27, 24]. Analogously, learngene mutation also operates at the structural level by modifying the arrangement of filters and kernels, to enhance adaptability for encoding core knowledge.

Given the structural alignment in Eq. (4), mutations are applied primarily to the filter sets $\mathcal{F}_l$ at each layer, with corresponding kernel indices $\mathcal{K}_{l+1}$ updated automatically to maintain connectivity. For each learngene, structural mutation is performed independently at each layer with probability $p_m$. The probabilities of adding ($p^+$) or removing ($p^-$) filters in the $l$-th layer of learngenes are given by:

$$p_l^- = \alpha \cdot \frac{|\mathcal{F}_l|}{N_F^{(l)} - |\mathcal{F}_l|}, \quad p_l^+ = 1 - p_l^+ \tag{5}$$

where $\alpha$ is a balancing coefficient and $|\cdot|$ denotes set cardinality. Mutations proceed layer-wise and may involve multiple filters per layer. The complete mutation procedure is detailed in Algorithm 1.

### 3.2.3 Inheritance of Learngenes

Learngene inheritance facilitates the transfer of core knowledge from a source model to target models with varying depths, widths, or architectures. To accommodate architectural differences during transfer, ECO employs the following strategies:

- *Zero-padding Incomplete Filters.* Since learngenes may include only a subset of kernels within each filter, direct insertion into wider filters can introduce noisy, randomly initialized parameters. To avoid disrupting the encoded knowledge, missing kernels are initialized as zeros, preserving functionality while enabling later adaptation (see Appendix Figure 8).

- *Index Reordering for Narrower Networks.* When transferring to narrower networks, mismatches in filter and kernel indices may arise. ECO resolves this by reordering indices according to their relative positions (see Appendix Figure 9), maintaining structural consistency and ensuring compliance with the alignment constraint in Eq. (4).

Additional inheritance strategies, including the proposed *partial identity mapping* for depth expansion, are detailed in Appendix B.4.

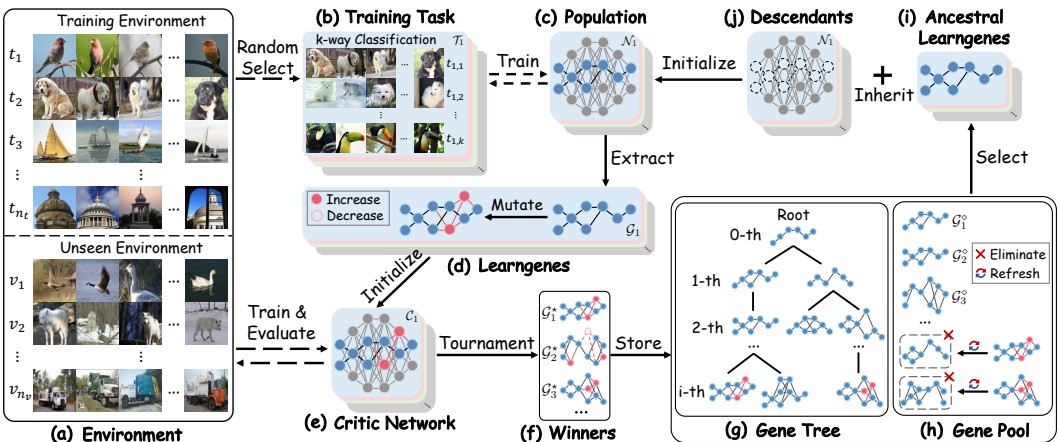

Figure 3: The Genetic Transfer Learning (GTL) framework. GTL iteratively condenses knowledge by training populations (**c**) on randomly sampled tasks (**a**, **b**) and selecting transferable core knowledge through mutation (**d**) and tournaments (**f**). The Gene Pool (**h**) and Gene Tree (**g**) store superior learngenes and track their kinship for inheritance (**i**, **j**), respectively.

## 3.3 Evolution for Learngene Extraction

Genetic Transfer Learning (GTL) is an evolutionary framework derived from GRL [11] that iteratively optimizes learngenes by transferring them across generations of neural networks. Through this inheritance mechanism, GTL progressively condenses and refines core knowledge into compact, reusable learngenes that generalize across diverse models and tasks, as illustrated in Figure 3 and formalized in Algorithm 2.

Briefly, each generation begins with a population of $n_p$ neural networks, each inheriting learngenes from the previous generation and trained on a randomly assigned task. After training, learngenes are extracted and undergo structural mutations to introduce diversity. A tournament-based selection mechanism is then employed to identify high-performing learngenes, which are retained in the Gene Pool to guide inheritance in the next generation. This iterative process of inheritance, mutation, and selection continues across generations, progressively refining the quality of learngenes. The core components of GTL are detailed below.

**(1) Training the Population of Neural Networks.** Let $\mathcal{D} = \mathcal{D}_{\text{train}} + \mathcal{D}_{\text{val}}$ be the dataset with $n_t$ training classes and $n_v$ validation classes. In each generation, a population $\mathcal{P} = \{\mathcal{N}_1, \mathcal{N}_2, ..., \mathcal{N}_{n_p}\}$ is created. Each network $\mathcal{N}_i$ randomly samples $k$ classes from $\mathcal{D}_{\text{train}}$ to form a $k$-way classification task $\mathcal{T}_i$, encouraging diverse learning environments (Figure 3a–c). As evolution progresses, $k$ increases to simulate growing task complexity, emulating evolutionary pressure[41, 26].

**(2) Selecting Superior Learngenes.** After training, each neural network $\mathcal{N}_i$ has updated its inherited learngene $\mathcal{G}_i$ by encoding task-specific experience, thereby refining the core knowledge. To promote adaptability and structural diversity, each $\mathcal{G}_i$ then undergoes structural mutation (Figure 3d and Section 3.2.2), introducing controlled variations that further shape its knowledge representation.

Each mutated learngene $\mathcal{G}_i$ is evaluated by initializing a critic network $\mathcal{C}_i$, which is trained on the validation set $\mathcal{D}_{\text{val}}$ and the resulting accuracy defines the score $s_i$ for $\mathcal{G}_i$ (Figure 3e). To select high-quality yet diverse learngenes, tournament selection is employed: in each round, $\epsilon$ learngenes are randomly sampled, and the one with the highest score is added to the winner set $\mathcal{G}^\star = \{\mathcal{G}_1^\star, \mathcal{G}_2^\star, \ldots, \mathcal{G}_{n_w}^\star\}$ (Figure 3f), where $n_w = \lceil \frac{n_p}{\epsilon} \rceil$ denotes the number of tournaments per generation.

**(3) Storing Superior Learngenes and Tracking Kinship.** Following tournament selection, the winner set $\mathcal{G}^\star$ is incorporated into the Gene Pool (GP), which retains high-quality learngenes for inheritance in future generations (Figure 3h). Initialized with the top-performing learngenes from the first generation, the GP maintains up to $\rho$ entries, denoted as GP = $\{\mathcal{G}_1^\diamond, \mathcal{G}_2^\diamond, \ldots, \mathcal{G}_\rho^\diamond\}$. In each generation, a subset of $\varepsilon$ learngenes from $\mathcal{G}^\star$ is admitted into the GP, replacing the lowest-performing entries to ensure both knowledge retention and evolutionary adaptability.

To trace inheritance across generations, GTL constructs a Gene Tree (GT) that records the evolutionary lineage of selected learngenes (Figure 3g). Each node in GT represents a superior learngene, with root nodes originating from the initial generation. New entries in GP are appended as leaf nodes, and the path length between nodes encodes their degree of kinship.

**(4) Updating Learngene Scores.** To preserve ancestral excellence and guide future evolution, the scores of learngenes in the GP are updated after each generation. For every selected learngene $\mathcal{G}_i^\star$ with score $s_i^\star$, its kinship is traced back from its corresponding leaf node to the root according to GT. Each ancestral learngene $\mathcal{G}_{\text{anc}}$ along this path receives a score update:

$$s_{\text{anc}} \leftarrow s_{\text{anc}} + \eta^\tau s_i^\star \tag{6}$$

where $\eta$ is the decay coefficient and $\tau$ is the path length. See Algorithm 3 for more details.

**(5) Generating the Next Generation of Learngenes.** After refreshing the GP and updating learngene scores, the next generation of learngenes is sampled from the GP according to a score-proportional probability:

$$p_i = \frac{s_i^\diamond}{\sum_{i=1}^{\rho} s_i^\diamond} \tag{7}$$

where $p_i$ is the probability of selecting $\mathcal{G}_i^\diamond$ as a parent based on its score $s_i^\diamond$ (Figure3i).

The selected learngenes are then inherited by a new population of networks $\mathcal{P}$ (see Figure 3j and Section 3.2.3), initiating the next cycle of evolution.

# 4 Experiments

**Datasets.** We conduct evolutionary experiments on three datasets of increasing scale. CIFAR-FS [2] and *mini*ImageNet [64] each contain 100 classes, split into 64 for training ($\mathcal{D}_{\text{train}}$), 16 for validation ($\mathcal{D}_{\text{val}}$), and 20 for novel evaluation. ImageNet-1K [10] contains 1,000 classes, divided into 640, 160, and 200 for the same purposes. We further evaluate the extracted learngenes on four downstream datasets: Oxford Flowers [46], CUB-200-2011 [66], Stanford Cars [15], and Food-101 [4].

**Network Architectures.** We evaluate four representative convolutional architectures: VGG11 [52], ResNet12 [20], ResNet50 [20], and MobileNetV3-Large [21]. To assess the scalability and adaptability of learngenes in initializing models of varying capacity, we additionally evaluate width-adjusted variants of ResNet50 and MobileNetV3-Large, with widths scaled from $0.5W$ to $0.9W$, where $W$ denotes the original width.

**Training Details.** Evolutionary training is conducted independently across networks to support parallelism. For VGG11 and ResNet12, learngenes evolve over 250 generations, each comprising 20 networks trained for 15 epochs. For MobileNetV3-Large and ResNet50, evolution proceeds for 100 generations, with 6 networks per generation trained for 5 epochs. All experiments are executed on NVIDIA GeForce RTX 4090 GPUs, with total computational cost comparable to training a typical medium-scale model. Full hyperparameter configurations are detailed in Appendix C.1.

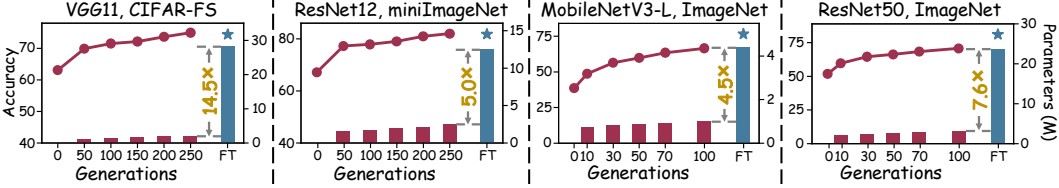

Figure 4: Accuracy (red curves) of networks on validation classes and parameter (red bars) of corresponding learngenes during evolution. Blue stars and bars represent performance and parameter transfer via direct fine-tuning. Notably, models inheriting learngenes achieve comparable or superior results, especially in smaller networks, with a significant reduction in transferred parameters.

# 5 Results

## 5.1 Evolutionary Performance of Learngenes Extracted by ECO

In biological evolution, beneficial mutations accumulate through natural selection, leading to the progressive refinement of genes [24, 28]. Similarly, ECO evolves learngenes by gradually refining core knowledge across generations.

As shown in Figure 4, learngenes accumulate dominant mutations, reflected in the steady growth of their parameter counts. Consequently, networks initialized with evolved learngenes demonstrate consistently improved performance over time. Notably, ECO significantly outperforms models trained from scratch, achieving performance comparable to pre-trained models while transferring significantly fewer parameters (14.5× reduction in VGG11).

Table 1 provides a quantitative comparison across generations against baseline methods. Taking ResNet12 on *mini*ImageNet as an illustrative example, early-generation learngenes already outperform direct initialization approaches (e.g., GradInit [85]) and hypernetwork-based methods (e.g., GHN-3 [30]), benefiting from inherited transferable knowledge. At mid-generation, ECO surpasses advanced transfer methods such as Heur-LG [68] and Auto-LG [70], transferring 4.8× fewer parameters—highlighting its advantage in preserving convolutional feature hierarchies.

In later stages, ECO achieves a 2.6 performance gain over re-initialization methods like KE [59], while reducing training time by 3×. Compared to pre-training-based approaches (e.g., Weight Selection [80]), ECO reduces parameter transfer over 5.0×, with performance gains of 3.85, demonstrating both efficiency and the superior transferability of its core knowledge.

Table 1: Performance of networks inheriting learngenes on validation and novelty classes across various datasets. Para.(M) represents the number of transferred parameters. N/A denotes that the criterion is not applicable to the respective method.

| | VGG, 28.18M | | | ResNet12, 12.44M | | |
|---|---|---|---|---|---|---|
| Dataset | CIFAR FS | | | *mini*ImageNet | | |
| Methods | Para. | Valid | Novel | Para. | Valid | Novel |
| He-Init [19] | 0 | 63.13 | 66.05 | 0 | 67.19 | 65.55 |
| GradInit [85] | N/A | 65.19 | 66.80 | N/A | 67.75 | 68.90 |
| ECO-10th | 0.83 | **65.44** | **67.75** | 1.17 | **73.31** | **72.80** |
| GHN-2 [29] | N/A | 59.63 | 62.10 | N/A | 68.56 | 66.05 |
| GHN-3 [30] | N/A | 57.56 | 61.45 | N/A | 77.56 | 77.55 |
| ECO-50th | 1.08 | 69.94 | 71.70 | 1.47 | 77.25 | 77.45 |
| ECO-100th | 1.30 | **71.56** | **73.25** | 1.71 | **77.88** | **78.15** |
| Heur-LG [68] | 7.09 | 65.25 | 68.35 | 9.43 | 70.75 | 69.45 |
| Auto-LG [70] | 9.19 | 68.00 | 71.15 | 9.51 | 75.25 | 75.55 |
| ECO-150th | 1.51 | **72.19** | **74.50** | 1.96 | **79.00** | **78.90** |
| KE$_{N_3}$ [59] | N/A | 69.81 | 74.10 | N/A | 75.94 | 77.60 |
| ECO-200th | 1.81 | **73.69** | **76.30** | 2.09 | **80.94** | **80.20** |
| Wt Select [80] | 28.09 | 63.31 | 65.75 | 12.41 | 76.81 | 77.75 |
| ECO-250th | 1.94 | **75.00** | **76.95** | 2.47 | **82.00** | **81.60** |
| | Mobile-L, 4.37M | | | ResNet50, 23.66M | | |
| Dataset | ImageNet | | | ImageNet | | |
| Methods | Para. | Valid | Novel | Para. | Valid | Novel |
| He-Init [19] | 0 | 38.54 | 38.70 | 0 | 51.85 | 53.48 |
| GradInit [85] | N/A | 41.86 | 41.13 | N/A | 52.06 | 51.94 |
| ECO-10th | 0.74 | **48.66** | **49.21** | 2.22 | **59.70** | **60.05** |
| GHN-2 [29] | N/A | 31.93 | 30.72 | N/A | 47.30 | 46.27 |
| GHN-3 [30] | N/A | 37.28 | 37.18 | N/A | 54.13 | 52.89 |
| ECO-30th | 0.82 | **56.41** | **55.28** | 2.47 | **64.54** | **64.99** |
| Heur-LG [68] | 1.76 | 43.55 | 43.07 | 14.99 | 52.89 | 54.54 |
| Auto-LG [70] | 1.77 | 50.76 | 50.13 | 15.21 | 55.10 | 55.60 |
| ECO-50th | 0.86 | **59.81** | **58.60** | 2.63 | **66.29** | **66.32** |
| KE$_{N_3}$ [59] | N/A | 54.58 | 54.52 | N/A | 65.93 | 67.02 |
| ECO-70th | 0.92 | **63.39** | **62.08** | 2.82 | **68.35** | **67.92** |
| Wt Select [80] | 2.94 | 56.03 | 55.84 | 23.45 | 69.91 | 69.29 |
| ECO-100th | 0.98 | **66.50** | **65.22** | 3.13 | **70.73** | **69.91** |

## 5.2 Performance of ECO in Initializing Models of Variable Sizes

Neural networks deployed in diverse hardware environments often require models of varying sizes due to differing computational and storage constraints, but pre-trained models are typically fixed in size, making it impractical to have one for every possible configuration.

ECO addresses this challenge through scalable and size-agnostic learngenes, enabling efficient initialization of networks with variable sizes. As shown in Table 2, ECO consistently achieves superior accuracy across variable-sized models, significantly outperforming existing transfer methods.

Compared to direct pre-training (i.e., Direct PT), ECO reduces training time by at least 2×, while transferring up to 2.5× and 3.7× fewer parameters than Weight Selection [80], highlighting its transfer efficiency. Moreover, ECO achieves a 3× reduction in FLOPs relative to STKD [57]. Unlike distillation-based approaches whose computational cost scales linearly with the number of target models ($O(n)$), ECO directly initializes each model independently with no additional training overhead, enabling scalable deployment across diverse resource constraints.

Table 2: Performance of ECO and other methods for variable-sized model initialization. "Com." refers to the complexity of initializing $n$ models, while "Time", "Para.(M)", and "FLOPs(G)" represent training time, transferred parameters, and computational overhead, respectively. N/A denotes that the criterion is not applicable to the respective method.

| | | | MobileNetV3-Large, 5.04M, ImageNet, Train | | | | | | | | | | | | | | |
| | | | 0.70W | | | 0.75W | | | 0.80W | | | 0.85W | | | 0.90W | | |
| Methods | Com. | Time | Para. | FLOPs | Acc. | Para. | FLOPs | Acc. | Para. | FLOPs | Acc. | Para. | FLOPs | Acc. | Para. | FLOPs | Acc. |
|---|---|---|---|---|---|---|---|---|---|---|---|---|---|---|---|---|---|
| He-Init [19] | O(1) | 1.0× | 0 | 0.75 | 32.71 | 0 | 0.87 | 34.48 | 0 | 0.92 | 34.54 | 0 | 1.04 | 36.13 | 0 | 1.15 | 37.07 |
| GradInit [85] | O(n) | 1.0× | N/A | 0.75 | 37.07 | N/A | 0.87 | 38.26 | N/A | 0.92 | 35.68 | N/A | 1.04 | 40.01 | N/A | 1.15 | 40.58 |
| Direct PT | O(n) | 2.0× | 0 | 0.75 | 42.35 | 0 | 0.87 | 44.61 | 0 | 0.92 | 44.52 | 0 | 1.04 | 46.54 | 0 | 1.15 | 47.97 |
| Wt Select [80] | O(1) | 1.0× | 1.48 | 0.75 | 42.87 | 1.70 | 0.87 | 44.83 | 1.90 | 0.92 | 45.30 | 2.16 | 1.04 | 46.88 | 2.41 | 1.15 | 47.85 |
| STKD [57] | O(n) | 1.0× | N/A | 2.14 | 46.03 | N/A | 2.40 | 46.78 | N/A | 2.51 | 45.13 | N/A | 2.78 | 47.66 | N/A | 3.02 | 48.08 |
| ECO | O(1) | 1.0× | 0.98 | 0.75 | **46.07** | 0.98 | 0.87 | **48.09** | 0.98 | 0.92 | **48.15** | 0.98 | 1.04 | **48.73** | 0.98 | 1.15 | **49.13** |
| | | | ResNet50, 24.87M, ImageNet, Train | | | | | | | | | | | | | | |
| | | | 0.50W | | | 0.55W | | | 0.60W | | | 0.65W | | | 0.70W | | |
| Methods | Com. | Time | Para. | FLOPs | Acc. | Para. | FLOPs | Acc. | Para. | FLOPs | Acc. | Para. | FLOPs | Acc. | Para. | FLOPs | Acc. |
| He-Init [19] | O(1) | 1.0× | 0 | 6.82 | 39.26 | 0 | 8.00 | 41.32 | 0 | 9.33 | 42.68 | 0 | 10.84 | 44.31 | 0 | 12.40 | 45.68 |
| GradInit [85] | O(n) | 1.0× | N/A | 6.82 | 37.28 | N/A | 8.00 | 39.28 | N/A | 9.33 | 41.09 | N/A | 10.84 | 42.68 | N/A | 12.40 | 43.97 |
| Direct PT | O(n) | 2.0× | 0 | 6.82 | 49.85 | 0 | 8.00 | 51.66 | 0 | 9.33 | 53.59 | 0 | 10.84 | 54.54 | 0 | 12.40 | 55.89 |
| Wt Select [80] | O(1) | 1.0× | 5.88 | 6.82 | 49.97 | 7.06 | 8.00 | 51.21 | 8.41 | 9.33 | 52.86 | 9.87 | 10.84 | 54.23 | 11.46 | 12.40 | 55.89 |
| STKD [57] | O(n) | 1.0× | N/A | 23.16 | 51.86 | N/A | 25.78 | 53.00 | N/A | 28.76 | 52.91 | N/A | 32.09 | 52.99 | N/A | 35.58 | 54.39 |
| ECO | O(1) | 1.0× | 3.13 | 6.82 | **52.23** | 3.13 | 8.00 | **53.12** | 3.13 | 9.33 | **53.79** | 3.13 | 10.84 | **54.88** | 3.13 | 12.40 | **55.93** |

Table 3: Performance of ECO and other methods when transferring to downstream datasets. "Para." refers to the transferred parameters, with total parameters and FLOPs recorded for various architectures. N/A denotes that the criterion is not applicable to the respective method.

| Methods | Para. | Flower | CUB | Cars | Food | *Aver.* | Para. | Flower | CUB | Cars | Food | *Aver.* |
|---|---|---|---|---|---|---|---|---|---|---|---|---|
| | **VGG11, FLOPs=45.14G, Para=28.72M** | | | | | | **ResNet12, FLOPs=154.04G, Para=12.52M** | | | | | |
| He-Init [19] | 0 | 34.22 | 49.81 | 60.94 | 78.67 | *55.91* | 0 | 50.74 | 46.89 | 61.27 | 81.10 | *60.00* |
| GradInit [85] | N/A | 36.38 | 49.36 | 66.86 | 78.07 | *57.67* | N/A | 50.92 | 55.61 | 72.45 | 81.39 | *65.09* |
| GHN-2 [29] | N/A | 46.32 | 49.05 | 68.18 | 67.57 | *57.78* | N/A | 54.92 | 49.36 | 59.21 | 74.10 | *59.40* |
| GHN-3 [30] | N/A | 44.09 | 50.55 | 69.39 | 66.49 | *57.63* | N/A | 61.33 | 56.06 | 71.30 | 75.28 | *65.99* |
| Heur-LG [68] | 7.09 | 38.84 | 55.54 | 71.67 | 78.65 | *61.18* | 9.43 | 54.69 | 55.68 | 76.37 | 81.93 | *67.17* |
| Auto-LG [70] | 9.19 | 50.64 | 58.54 | 74.93 | 78.60 | *65.68* | 9.51 | 61.90 | 58.99 | 81.68 | 82.15 | *71.18* |
| Wt Select [80] | 28.09 | 62.87 | 60.08 | 76.21 | 78.26 | *69.36* | 12.41 | 55.26 | 57.92 | 77.74 | 81.68 | *68.15* |
| ECO | 1.94 | **64.42** | **60.20** | **78.12** | **79.47** | *70.55* | 2.47 | **81.48** | **64.14** | **84.28** | **82.41** | *78.08* |
| | **MobileNetV3-Large, FLOPs=1.33G, Para=4.39M** | | | | | | **ResNet50, FLOPs=24.62G, Para=23.81M** | | | | | |
| He-Init [8] | 0 | 51.05 | 56.08 | 74.88 | 74.81 | *64.21* | 0 | 27.53 | 46.34 | 46.96 | 73.66 | *48.62* |
| GradInit [85] | N/A | 56.38 | 56.33 | 71.67 | 54.49 | *59.72* | N/A | 46.09 | 48.15 | 53.84 | 75.48 | *55.89* |
| GHN-2 [29] | N/A | 46.07 | 47.53 | 56.96 | 66.35 | *54.23* | N/A | 61.68 | 55.66 | 66.91 | 66.79 | *62.76* |
| GHN-3 [30] | N/A | 42.75 | 45.06 | 54.40 | 58.54 | *50.19* | N/A | 49.18 | 52.69 | 71.58 | 71.26 | *61.18* |
| Heur-LG [68] | 1.76 | 56.12 | 58.23 | 75.43 | 74.31 | *66.02* | 14.99 | 53.78 | 54.95 | 67.08 | 72.83 | *62.16* |
| Auto-LG [70] | 1.77 | 56.77 | 59.15 | 77.98 | 74.69 | *67.15* | 15.21 | 55.42 | 57.11 | 71.14 | 73.28 | *64.24* |
| Wt Select [80] | 2.94 | 61.81 | 61.56 | 79.17 | 74.77 | *69.33* | 23.45 | 57.16 | 52.42 | 62.58 | 75.54 | *61.93* |
| ECO | 0.98 | **64.45** | **62.17** | **79.90** | **75.04** | *70.39* | 3.13 | **77.02** | **65.12** | **82.61** | **76.13** | *75.22* |

## 5.3 Performance of ECO on Downstream Tasks

The core knowledge encapsulated in learngenes exhibits strong generalizability, enabling effective transfer across a wide range of downstream tasks. As shown in Table 3, ECO consistently outperforms baseline methods, confirming its robust adaptability and transfer efficiency.

Hypernetwork-based approaches (e.g., GHN-2 [29], GHN-3 [30]) achieve competitive results on small datasets but often underperform on larger ones due to their coarse architecture-level parameter modeling. Methods like Heur-LG [68] and Auto-LG [70] transfer pre-trained knowledge layer-wise, but this localized mapping can disrupt global feature hierarchies, especially in compact architectures. Notably, Auto-LG lags behind ECO by 4.87 on average with VGG11, despite transferring 4.7× more parameters, reflecting inefficiencies in knowledge utilization.

Pre-trained model-based approaches offer extensive knowledge transfer, but their reliance on task-agnostic optimization can lead to suboptimal adaptation. On Food-101 with VGG11, Weight Selection achieves 78.26, in contrast to ECO's 79.47, highlighting the need for more targeted transfer mechanisms. These results underscore ECO's ability to retain and apply transferable core knowledge more efficiently, offering a scalable and task-adaptive solution for downstream model initialization.

Table 4: Accuracy of few-shot classification. "-N" indicates narrower networks than normal ones.

| Methods | CIFAR-FS, VGG11 | | | *mini*Imagenet, ResNet12 | | |
|---|---|---|---|---|---|---|
| | 5-shot | 10-shot | 20-shot | 5-shot | 10-shot | 20-shot |
| MAML[14] | 63.4±0.86 | 68.2±0.74 | 70.5±0.77 | 61.1±0.78 | 66.4±0.68 | 68.4±0.62 |
| RelationNet[58] | 64.2±0.79 | 68.9±0.71 | 72.9±0.71 | 65.4±0.69 | 70.3±0.66 | 72.9±0.63 |
| MatchingNet[64] | 59.9±0.78 | 63.8±0.78 | 69.3±0.81 | 66.3±0.66 | 70.9±0.63 | 74.7±0.59 |
| ProtoNet[54] | 65.9±0.85 | 69.3±0.79 | 73.1±0.69 | 66.5±0.71 | 72.4±0.60 | 74.9±0.59 |
| Baseline++[7] | 64.9±0.78 | 71.3±0.73 | 75.3±0.67 | 67.5±0.67 | 74.0±0.60 | 78.2±0.51 |
| ECO | 69.9±0.78* | 75.5±0.69* | 78.5±0.63* | 69.4±0.71* | 75.4±0.61* | 80.2±0.52* |
| ECO-N | **70.5±0.73** | **76.6±0.65** | **80.5±0.58** | **71.3±0.70** | **76.8±0.59** | **81.7±0.53** |

Table 5: Ablation study results across various architectures.

| Methods | VGG11 | | ResNet12 | | MobileNetV3-L | | ResNet50 | |
|---|---|---|---|---|---|---|---|---|
| | Valid | Novel | Valid | Novel | Valid | Novel | Valid | Novel |
| He Init [19] | 63.13 | 66.05 | 67.19 | 65.55 | 38.54 | 38.70 | 51.85 | 53.48 |
| Direct Select | 70.06 | 72.30 | 77.50 | 76.85 | 52.46 | 52.14 | 60.63 | 61.23 |
| w/o Tournament & GP | 72.31 | 74.55 | 80.94 | 80.25 | 62.06 | 60.79 | 65.79 | 65.97 |
| w/o Mutation | 72.25 | 73.25 | 80.88 | 79.80 | 63.08 | 62.11 | 69.00 | 68.72 |
| ECO | **75.00** | **76.95** | **82.00** | **81.60** | **66.50** | **65.22** | **70.73** | **69.91** |

## 5.4 Improved Data Efficiency of ECO

Models initialized with learngenes demonstrate strong data efficiency, particularly in few-shot learning scenarios. As shown in Table 4, ECO consistently outperforms conventional few-shot learning methods, including those built upon fully pre-trained models such as Baseline++ [7], underscoring the effectiveness of core knowledge encapsulated in learngenes under limited data conditions.

Notably, a narrower variant ("-N"), initialized with the same learngenes, achieves even higher accuracy. This can be attributed to reduced parameter redundancy and stochasticity, indicating that learngenes in ECO not only encapsulate transferable representations but also enable more stable and efficient adaptation under both architectural and data constraints, thereby reinforcing its practicality and robustness in real-world, resource-constrained scenarios.

## 5.5 Ablation and Analysis

### 5.5.1 Effects of Mutation

Mutations are essential to the selection and refinement of core knowledge, as they dynamically reshape the structure and semantics of learngenes. By introducing controlled variations, mutations promote population diversity and enhance the effectiveness of tournament-based selection.

As illustrated in Figure 5, learngenes within ResNet12 undergo continual structural evolution, progressively condensing transferable knowledge while eliminating redundant components, facilitating the emergence of more generalizable representations. Thus, the absence of mutation leads to static learngene structures, which constrain learngene diversity and hinder knowledge accumulation, as demonstrated in Table 5.

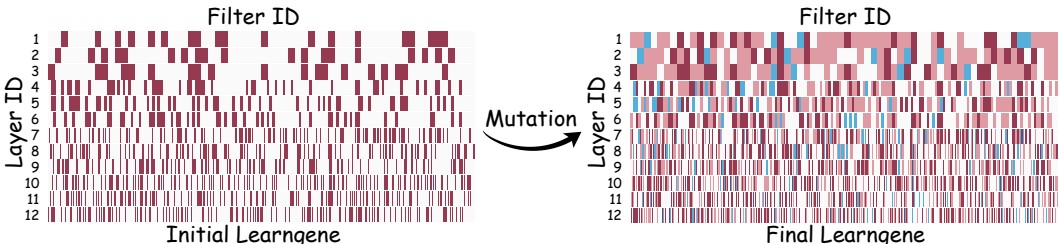

Figure 5: Visualization of learngenes in ResNet12 pre- and post-evolution. ■ is the filter of the initial learngenes. ■ is the filter that becomes a new part of learngene through mutation. ■ indicates redundant filters removed during the evolution.

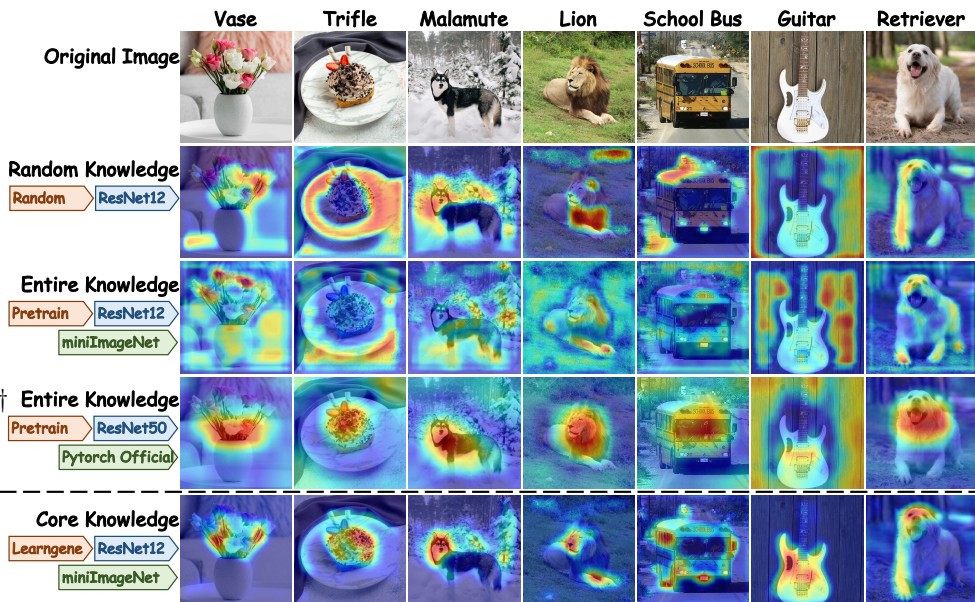

Figure 6: Visualization of core knowledge in learngenes. Networks have not undergone any learning or fine-tuning. † ResNet50 (pre-trained on ImageNet) already contains these classes.

### 5.5.2 Effects of Evolution

ECO leverages evolution to extract and refine core knowledge, enhancing transferability across model sizes and tasks. In contrast, traditional pre-trained models are optimized for specific objectives and lack mechanisms for isolating generalizable components. As a result, methods that directly extract fragments from such models (i.e., Direct Select) fail to capture broadly applicable knowledge, as shown in Table 5.

Tournament selection plays a critical role in identifying high-quality learngenes for preservation in the Gene Pool (GP), thereby facilitating the accumulation and iterative refinement of transferable core knowledge across generations. As evidenced in Table 5, the removal of tournament selection and GP (i.e., w/o Tournament & GP) disrupts this process, resulting in the retention of redundant or suboptimal knowledge and weakening the influence of superior candidates across generations.

### 5.6 Visualization of Core Knowledge in Learngenes

To illustrate the core knowledge encapsulated in learngenes, we visualize model attention using CAM [51] on sample images from novel classes in *mini*ImageNet, which are not involved during learngene evolution.

As shown in Figure 6, randomly initialized models tend to focus on diffuse or irrelevant regions, while pre-trained models, though more focused, often highlight background areas. For instance, pre-trained ResNet12 fails to localize novel objects, and ResNet50, despite identifying relevant regions, exhibits background activation that may introduce bias (see Figure 1a).

In contrast, models initialized with learngenes produce compact, focused attention maps, concentrating on semantically meaningful and discriminative regions, even to unseen categories.

## 6 Conclusion

Inspired by biological knowledge transfer, we propose ECO, a method that condenses core knowledge into learngenes for efficient transfer across models. Built upon the Genetic Transfer Learning (GTL) framework, ECO enables the evolution of neural networks and the inheritance of learngenes in supervised tasks. Experiments show that learngenes impart strong learning capabilities, while ensuring scalability and adaptability across model sizes and tasks, providing an efficient and generalizable alternative for knowledge transfer through the inheritance of core knowledge.

## Acknowledgement

We sincerely appreciate Freepik for contributing to the figure design. This research was supported by the Jiangsu Science Foundation (BG2024036, BK20243012), the National Natural Science Foundation of China (62125602, U24A20324, 92464301, 62306073), China Postdoctoral Science Foundation (2022M720028, 2025T180432), the Xplorer Prize, the Fundamental Research Funds for the Central Universities (2242025K30024), and SEU Innovation Capability Enhancement Plan for Doctoral Students (CXJH_SEU 26023).

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

# A  Additional Related Work

## A.1  Common Knowledge in Neural Networks

Common knowledge refers to transferable representations that can be shared across neural networks. It typically falls into two categories: size-agnostic knowledge, which generalizes across network architectures of different scales, and task-agnostic knowledge, which transfers across tasks.

**Size-agnostic Knowledge.** The modular nature of neural networks (e.g., convolutional filters and Transformer blocks) enables knowledge sharing across models of varying widths and depths. For example, Filter-in-Filter [76], versatile filters [72], and TBC [71] show that filters can be reused to detect multiple patterns or reduce redundancy. Other works [36, 81, 34] compress CNNs through cross-filter weight sharing. Similarly, in Transformer-based models, reusable blocks [75] and parameter-sharing techniques [33, 83] achieve efficient scaling without performance degradation.

**Task-agnostic Knowledge.** Task-agnostic knowledge captures general visual features that are transferable across domains. Tape [38] encodes such priors for image restoration, while Park et al.[47] and Polyhistor[39] learn generalizable representations and reusable adapters for multi-task learning. The idea of a universal backbone, as in Universal Template [62], improves generalization in few-shot settings. Meta-learning approaches like OML [25] and iTAML [48] enhance transferability by decoupling task-specific heads from shared representations.

ECO integrates and condenses such size- and task-agnostic knowledge into adaptive neural fragments called learngenes, which encapsulate what we refer to as core knowledge. These learngenes enable flexible adaptation across network scales and task types.

# B  Additional Details of Methods

## B.1  Learngenes in ResNets

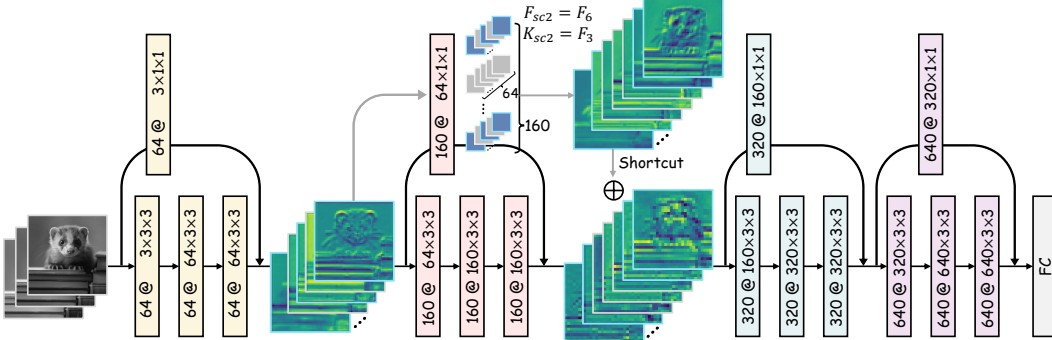

Figure 7: The form of the learngenes in ResNet12, where the kernels in skip connection layers are also integrated as components of the learngenes.

ResNets enhance traditional CNN architectures by introducing skip connections, typically realized through $1\times1$ convolutions. These connections enable feature propagation across non-adjacent layers, promoting representational reuse.

To accommodate this architectural feature, the formulation of learngenes in ResNets should extended beyond standard convolutional layers to explicitly incorporate skip connections. Let $L_i$ and $L_j$ denote the source and target layers of a skip connection, where $L_j > L_i$, and let the corresponding skip layer be denoted as $L_{\mathrm{sc}}$. To ensure structural alignment with the main pathway, the number of kernels and filters in the learngenes at $L_{\mathrm{sc}}$ are defined as:

$$N_K^{(L_{\mathrm{sc}})} = N_F^{(L_i)}, \; N_F^{(L_{\mathrm{sc}})} = N_F^{(L_j)} \tag{8}$$

Accordingly, the selected kernel and filter index sets of learngenes are given by:

$$\mathcal{K}_{\mathrm{sc}} = \mathcal{F}_i, \; \mathcal{F}_{\mathrm{sc}} = \mathcal{F}_j \tag{9}$$

This configuration ensures that learngenes encapsulate both primary and residual pathways, forming complete kernel-level circuits essential for effective knowledge transfer, as illustrated in Figure 7.

---

**Algorithm 1** Mutation of the Learngene

---

**Input**: Learngene $\mathcal{G}$, Mutation probability $p_m$, Number of layers $N_L$ and number of filters $N_F^{(l)}$ in $l$-th layer

1: *# Mutation layer by layer*
2: **for** $l = 1$ to $N_L$ **do**
3:     *# Whether to mutate in l-th layer*
4:     Randomly generate a number $r \sim U(0, 1)$.
5:     **while** $r \leq p_m$ **do**
6:        *# Mutate!*
7:        *# Whether to increase and decrease a filter of $\mathcal{G}$ in layer l*
8:        Randomly generate a number $s \sim U(0, 1)$.
9:        **if** $s \leq p_l^+$ **then**
10:           *# Random increase a filter*
11:           Randomly select a filter index $f$ from $[1, N_F^{(l)}] - \mathcal{F}_l$
12:           *# Update the corresponding sets $F_l$ and $K_l$*
13:           $\mathcal{F}_l \leftarrow \mathcal{F}_l \cup \{f\}$
14:           $\mathcal{K}_{l+1} \leftarrow \mathcal{K}_{l+1} \cup \{f\}$
15:        **else**
16:           *# Random decrease a filter*
17:           Randomly select a filter index $f$ from $\mathcal{F}_l$
18:           *# Update the corresponding sets $\mathcal{F}_l$ and $\mathcal{K}_l$*
19:           $\mathcal{F}_l \leftarrow \mathcal{F}_l - \{f\}$
20:           $\mathcal{K}_{l+1} \leftarrow \mathcal{K}_{l+1} - \{f\}$
21:        **end if**
22:        *# Whether to continue mutating in layer l*
23:        Randomly generate $r \sim U(0, 1)$.
24:     **end while**
25: **end for**

---

---

**Algorithm 2** Genetic Transfer Learning

---

**Input**: Training dataset $\mathcal{D}_{\text{train}}$ with $n_t$ classes, Validate dataset $\mathcal{D}_{\text{val}}$ with $n_v$ classes, Population number $n_p$, and total number of generation $N_G$

1: **for** $g = 0$ to $N_G$ **do**
2:     *# Initialize population with learngenes*
3:     Randomly initialize population $\mathcal{P}_g$ with $n_p$ networks $\mathcal{N}_i$
4:     **if** $g \neq 0$ **then**
5:        Select ancestry learngenes $\mathcal{G} = \{\mathcal{G}_1, \mathcal{G}_2, ..., \mathcal{G}_{n_p}\}$ from the Gene Pool for each $\mathcal{N}_i$ using Eq. (7). Then initialize each $\mathcal{N}_i$ by inheriting $\mathcal{G}_i$
6:     **end if**
7:     *# Train population $\mathcal{P}$ on $\mathcal{D}_{train}$*
8:     **for** each network $\mathcal{N}_i$ **do**
9:        Sample $k$ classes from $\mathcal{D}_{\text{train}}$ to form task $\mathcal{T}_i$
10:        Train $\mathcal{N}_i$ on $\mathcal{T}_i$
11:     **end for**
12:     *# Learngene extraction and evaluation*
13:     **for** each learngene $\mathcal{G}_i$ **do**
14:        Mutate $\mathcal{G}_i$ according to Appendix Algorithm 1
15:        Initialize critic networks $\mathcal{C}_i$ with mutated $\mathcal{G}_i$
16:        Train $\mathcal{C}_i$ on a $n_v$-way classification task on $\mathcal{D}_{\text{val}}$
17:        Calculate learngene score $s_i$ based on the accuracy of $\mathcal{C}_i$
18:     **end for**
19:     *# Update Gene Pool and Gene Tree*
20:     Perform learngene competition and select winners $\mathcal{G}^\star$
21:     Take winners as leaf nodes of the Gene Tree
22:     Update scores of ancestral learngenes according to Eq. (6)
23:     Refresh Gene Pool by adding winners and eliminating learngenes with lower scores
24: **end for**

---

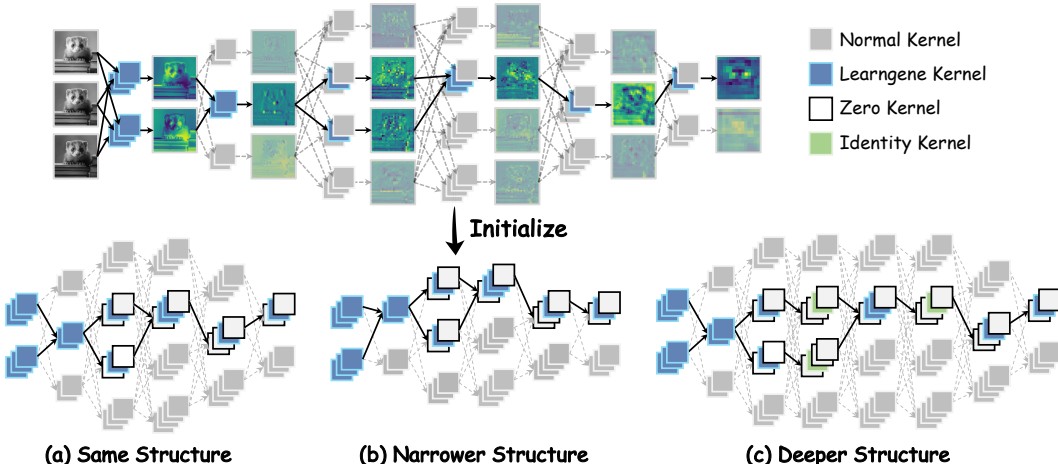

Figure 8: The learngenes exhibit scalability, enabling the initialization of networks with the same structures and the flexibility to initialize networks with narrower/wider and deeper structures. The normal kernels in networks are randomly initialized, while the zero kernels and identity kernels are initialed by $\mathbf{0}$ (i.e., zero matrix) and $\mathring{\mathbf{1}}$, respectively.

## B.2 Further Details on Learngene Mutation

Learngene mutation serves not only to alter structural configurations but also to refine the condensed core knowledge. This process increases structural diversity while maintaining a broad reservoir of "raw material" for subsequent selection, thereby facilitating the identification of core knowledge most suitable for transfer.

The complete mutation procedure is outlined in Algorithm 1.

## B.3 Further Details of Genetic Transfer Learning (GTL)

The complete GTL procedure is presented in Algorithm 2, complementing the illustration in Figure 3.

A key component of GTL is the recursive update of learngene scores, which begins at a selected high-performing learngene (i.e., a leaf node in the current generation) and proceeds upward along its lineage to the root node. This process ensures that the performance feedback of descendants is reflected in the evaluation of their ancestors. The specific score update strategy is described in Algorithm 3.

---

**Algorithm 3** Update of the Learngene Score

**Input**: Winner learngenes $\mathcal{G}^\star$, Parental decay coefficient $\eta$, Path length $\tau$ and Gene Tree (GT)

1: **for** each $\mathcal{G}_i^\star$ in $\mathcal{G}^\star$ **do**
2:     Initialize node pointer $pt \leftarrow \mathcal{G}_i^\star$
3:     Set path length $\tau \leftarrow 0$
4:     *# Recursively traverse Gene Tree for ancestral learngenes*
5:     **while** $pt$ has parent $\mathcal{G}_p$ **do**
6:         *# Increase path length*
7:         $\tau \leftarrow \tau + 1$
8:         *# Update parental learngene score $s_p$*
9:         $s_p \leftarrow s_p + \eta^\tau s_i^\star$
10:        *# Move pointer*
11:        $pt \leftarrow \mathcal{G}_p$
12:     **end while**
13: **end for**

---

### B.4 Further Details on Learngene Inheritance

Learngenes possess high structural scalability, enabling the initialization of target networks with varying widths, depths, and architectures, as illustrated in Figure 8.

**Identical Model Sizes and Architectures.** When the target network shares the same architecture as the source (Figure 8a), initialization is achieved by directly replacing the randomly initialized kernels at the indices specified by the learngenes. Filters containing unassigned kernels are padded with zeros, denoted as white kernels in Figure 8.

**Width Adaptation.** For target networks that differ in width, index mappings must be adapted to avoid index overflows. Specifically, the filter and kernel indices in the $l$-th layer are re-indexed as $\mathcal{F}_l' = [1, |\mathcal{F}_l|]$ and $\mathcal{K}_l' = [1, |\mathcal{K}_l|]$, ensuring compatibility with the narrower or wider target network (Figure 8b and Figure 9).

**Depth Extension.** When the number of layers in the target network $N_L^{(d)}$ exceeds that in the learngene $N_L^{(a)}$, additional *Partial Identity Mapping* (PIM) layers are inserted to preserve feature continuity (Figure 8c). Each PIM layer bridges adjacent layers $L_l$ and $L_{l+1}$ by introducing an intermediate mapping: $L_l \rightarrow L_{\text{pim}} \rightarrow L_{l+1}$, where the filter set is defined as:

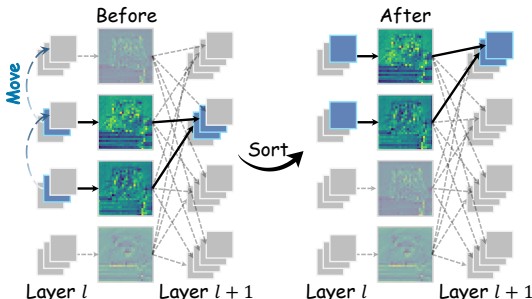

Figure 9: Sorting the positions of learngene filters and kernels. To prevent index overflow when initializing a narrower model, the indices of learngene kernels and filters are rearranged starting from 1.

$$\mathcal{F}_{\text{pim}} = \mathcal{F}_l \tag{10}$$

For each filter $f \in \mathcal{F}_l$, the corresponding kernel in the PIM layer is initialized as:

$$\mathcal{K}_{k,f,\text{pim}} = \begin{cases} \mathring{\mathbf{1}} & \text{if } k = f \\ \mathbf{0} & \text{otherwise} \end{cases} , \quad \mathring{\mathbf{1}}_{3\times3} = \begin{bmatrix} 0 & 0 & 0 \\ 0 & 1 & 0 \\ 0 & 0 & 0 \end{bmatrix} \tag{11}$$

This identity-preserving initialization ensures seamless propagation of core features across expanded network depths.

## C Implementation Details

### C.1 Hyperparameters for Evolution

Table 6 provide the hyperparameters used for evolving the learngenes using GTL.

Table 6: Hyperparameters in evolution.

| Hyperparameter | VGG11 | ResNet12 | ResNet50 | MobileNetV3-Large |
|---|---|---|---|---|
| Training Class Number $n_t$ | 64 | 64 | 640 | 640 |
| Validation Class Number $n_v$ | 16 | 16 | 160 | 160 |
| Novel Class Number | 20 | 20 | 200 | 200 |
| Init Class $k_{\text{init}}$ | 5 | 5 | 50 | 50 |
| Max Class $k_{\text{max}}$ | 15 | 15 | 70 | 110 |
| Initial Percentage of Learngene Filters | 0.3 | 0.3 | 0.3 | 0.4 |
| Mutation Probability $p_m$ | 0.1 | 0.1 | 0.2 | 0.2 |
| Population Size $n_p$ | 20 | 20 | 6 | 6 |
| Gene Pool Size $\rho$ | 8 | 8 | 6 | 6 |
| Obsolete Number $\varepsilon$ | 4 | 4 | 2 | 2 |
| Generational Decay Coefficient $\beta$ | 0.9 | 0.9 | 0.9 | 0.9 |
| Parental Decay Coefficient $\eta$ | 0.1 | 0.1 | 0.1 | 0.1 |
| Networks Number in Competition $s$ | 3 | 3 | 3 | 3 |
| Generation Number | 250 | 250 | 100 | 100 |

Table 7: Performance of ECO across various model sizes and downstream datasets. "Scratch" refers to training from scratch, while "$\mathcal{G}_{\text{vgg11}}$" and "$\mathcal{G}_{\text{res12}}$" represent the initialization using learngenes extracted from VGG11 and ResNet12, respectively. "-N/W" denotes narrower or wider network width. Note that results are given in terms of Top-3 accuracy.

| Datasets | | VGG | | | | | ResNet | | | |
| | | 11 | $11_{-N}$ | $11_{-W}$ | 16 | 19 | 12 | $12_{-N}$ | $12_{-W}$ | 18 |
|---|---|---|---|---|---|---|---|---|---|---|
| Flower | Scratch | 51.41 | 49.18 | 53.24 | 47.32 | 41.78 | 56.64 | 56.16 | 56.58 | 56.24 |
| | $\mathcal{G}_{\text{vgg11}}$ | 69.25* | 66.94* | 66.97* | 72.00* | 74.50* | 68.99 | 66.22 | 66.21 | 63.15 |
| | $\mathcal{G}_{\text{res12}}$ | 59.15 | 56.94 | 57.65 | 69.75 | 60.66 | 77.33* | **79.83*** | 74.63* | 76.96* |
| CUB | Scratch | 70.75 | 66.86 | 73.28 | 66.60 | 69.31 | 63.96 | 63.72 | 64.03 | 67.57 |
| | $\mathcal{G}_{\text{vgg11}}$ | 79.65* | 78.67* | 81.57* | 82.74 | **84.21*** | 76.67 | 75.15 | 76.70 | 78.24 |
| | $\mathcal{G}_{\text{res12}}$ | 77.89 | 75.99 | 79.58 | 83.53* | 81.15 | 81.55* | 82.24* | 80.34* | 83.62* |
| Cars | Scratch | 86.01 | 81.07 | 88.24 | 87.14 | 89.16 | 80.13 | 74.43 | 80.40 | 82.51 |
| | $\mathcal{G}_{\text{vgg11}}$ | 92.25* | 91.36* | 93.04* | 94.62 | 95.29* | 92.80 | 91.74 | 93.02 | 93.62 |
| | $\mathcal{G}_{\text{res12}}$ | 89.78 | 88.05 | 91.33 | 95.20* | 94.68 | 95.63* | 95.92* | 95.37* | **96.10*** |
| Food | Scratch | 80.74 | 79.73 | 80.29 | 79.54 | 76.23 | 84.62 | 84.21 | 85.28 | 87.59 |
| | $\mathcal{G}_{\text{vgg11}}$ | 85.49* | 84.73* | 84.79* | 87.30* | 86.85* | 88.19 | 87.32 | 88.32 | 89.00 |
| | $\mathcal{G}_{\text{res12}}$ | 84.69 | 83.06 | 84.44 | 86.46 | 84.28 | 89.54* | 89.09* | 89.74* | **90.57*** |

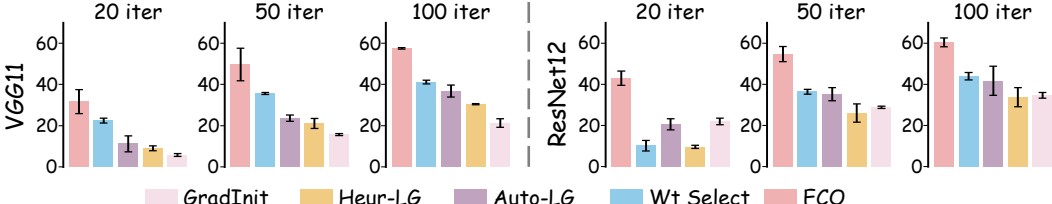

Figure 10: Performance of ECO and other methods on novelty classes with minimal training steps. "iter" indicates parameter update iterations (one iteration equals one optimizer update).

# D   Additional Results and Analysis

## D.1   Scalability and Adaptability of Learngenes

To further evaluate the scalability and adaptability of learngenes, we initialize networks with varying widths (e.g., VGG11-N, VGG11-W, ResNet12-N, ResNet12-W), depths (e.g., VGG16, VGG19, ResNet18), and even across architectures (e.g., VGG⇌ResNet). The results on four downstream datasets are presented in Table 7.

Learngenes (i.e., $\mathcal{G}_{\text{vgg11}}$ and $\mathcal{G}_{\text{res12}}$) successfully initialize networks of varying depths (e.g., VGG11, VGG16, VGG19; ResNet12, ResNet18), consistently outperforming models trained from scratch. They also enhance performance when initializing networks with different widths. For example, $\mathcal{G}_{\text{vgg11}}$ initialize a wider VGG11-W, achieving better performance on CUB (81.57% vs. 79.65%), while $\mathcal{G}_{\text{res12}}$ initialize a narrower ResNet12-N, surpassing the standard ResNet12 on Flower (79.83% vs. 77.33%).

Moreover, learngenes demonstrate effective cross-architecture transferability, with $\mathcal{G}_{\text{vgg11}}$ successfully initializing ResNet18 and outperforming models trained from scratch. Despite potential architectural incompatibilities, learngenes show greater flexibility and scalability compared to pre-trained models, which are often limited by architecture. This underscores their capability to transfer core knowledge across varying model sizes and architectures in downstream tasks.

## D.2   Instincts of Models Brought by Learngenes

To further explore the initialization ability of learngenes, we extend the concept of "instinct" from reinforcement learning [11] to supervised learning, referring to the innate ability that model initialization provides to neural networks. Figure 10 illustrates the early training performance (within the first epoch) of ECO and other methods on VGG11 and ResNet12, with one iteration represents a single optimizer step.

ECO exhibits notable advantages during early training, requiring only minimal updates to achieve significant performance improvements. Despite transferring more parameters, other methods often struggle to adapt quickly to new classes due to over-transfer of knowledge, which reduces network flexibility. In contrast, ECO selectively transfers only core knowledge via learngenes, enabling descendant networks to quickly adapt to novel tasks. This early-stage capacity for classification with minimal training is referred to as the "instinct" provided by learngenes.

## E   Limitations and Future Works

ECO demonstrates robust performance on convolutional architectures by integrating size- and task-agnostic knowledge through learngenes. While inherently scalable, current evaluations are limited to small and medium-sized models. Extending ECO to larger architectures remains promising, though it may incur additional training time and computational overhead. Beyond convolutional networks, we also aim to extend ECO to transformer-based models, where the modular representation of learngenes may further facilitate cross-architecture knowledge inheritance and generalization.

## F   Impact Statement

ECO offers a unified framework for modular knowledge transfer through learngenes, enabling efficient model initialization across architectures and scales. By reducing the need for extensive retraining, ECO lowers computational costs and improves performance in low-data and resource-constrained scenarios. Its architecture-agnostic design supports broad applicability across domains. By promoting the reuse of pre-trained models with minimal overhead, ECO contributes to more sustainable and accessible machine learning, aligning with the goals of green AI.

