# OpenReview forum: "ECO: Evolving Core Knowledge for Efficient Transfer"
_NeurIPS.cc/2025/Conference — NeurIPS 2025 poster_

### Official Review · Reviewer_X8DF · 2025-07-01

**Clarity:** 3
**Significance:** 3
**Originality:** 3
**Rating:** 5
**Confidence:** 2

**Summary:**

The authors introduce a new method for transfer learning in CNNs, ResNets, and vision transformers based on learngenes, neural circuits that are mutated and propagated through generations. In CNNs, learngenes are defined as collections of kernels that extend from network input to output, preserving information flow and facilitating possible transfer to other networks. In vision transformers, learngenes are realized as continuous circuits that extend from input to output. learngenes are inserted into networks and then trained on k-way image classification. They are then extracted, mutated, and inserted into critic networks, which evaluate learngenes on their ability to transfer core knowledge by performing classification on held out categories. Through tournament-style competition, the best learngenes are selected and passed on to the next generation. This method outperforms others based on gradient heuristics and meta-learning and is more efficient in terms of parameter transfer than fine-turning while retaining similar performance. ECO appears to shine in few-shot contexts. The authors claim their method leads vision transformers to generate focused attention maps when classifying unseen classes of objects.

**Questions:**

Major:

When is supervised learning performed in the training loop? Ideally this would be included in figure 3.

What was the rationale behind the decision to constrain the attention matrices as in eq. 12?

Why are learngenes structurally mutated before initializing the critic? The rationale for this design choice is unclear to me.

Minor:

Do the authors know what is being learned at the circuit level? A follow-up paper that determines exactly what ECO extracts would be of great interest.

**Ethical Concerns:**

["NO or VERY MINOR ethics concerns only"]

**Final Justification:**

The authors have adequately addressed my concerns. I trust that the final version of the paper will mention the follow up analysis on removing the structural constraint from learngene architecture and also mention that the validation classes are technically part of the training data. I will maintain my score.

**Limitations:**

Yes.

**Paper Formatting Concerns:**

No major paper formatting concerns.

**Quality:**

4

**Strengths And Weaknesses:**

Strengths:

This is a thorough and complete study that proves the utility and generalizability of the ECO method by testing across a range of architectures and datasets. Transfer learning as the inheritance of useful neural circuits appears to be effective and also comports heavily with what we know from biology. The authors demonstrate ECO can identify reusable motifs that can be inserted into architectures of varying sizes, retaining performance while transferring fewer parameters than other methods. The method appears to be particularly strong in the few-shot context. The paper is clear, demonstrates substantial knowledge of the relevant literature, and is easy to follow.

Weaknesses:

Overall, I think the paper had very few weaknesses.

My primary concern is that the validation classes are actually used to train the learngenes via the described method; therefore, scores on this division of the data are analogous to training and not test performance. This should be made clear to the reader.

Some details of the training regimen are a bit unclear, e.g. when typical backpropagation is used and for how many training iterations.

Also, it would be nice to show these results hold in the event that equation 12 in the appendix is relaxed, i.e. that attention heads do not have matching key, query, and value matrices.

The following papers should probably be cited:
Transfer Learning Layer Selection Using Genetic Algorithm

---

> ### Author Rebuttal · Authors · 2025-07-30
>
> Dear Reviewer X8DF,
>
> We sincerely appreciate your valuable feedback and the recognition of the innovation and performance demonstrated in our work. Your thoughtful and detailed review has been highly beneficial, and we are grateful for the time and effort you dedicated to evaluating our manuscript. Below, we provide our detailed responses to your comments and suggestions.
>
> ---
>
> > *Q1: **The validation classes are actually used to train the learngenes** via the described method; therefore, scores on this division of the data are analogous to training and not test performance*
>
> Thank you for your thoughtful feedback. To clarify, **validation classes are not used for training learngenes but rather for evaluating their generality during evolution**. This process is facilitated by a critic network, which inherits the learngene and is trained on the validation classes. However, **the updates to the critic network are discarded after each evaluation and do not impact the parameters of the learngenes being assessed**.
>
> Thus, **no knowledge from the validation classes is incorporated into the learngenes during evolution.**
> The results on validation classes presented in Table 1 are intended to show that learngenes perform equally well on both novel and validation classes, proving they don’t overfit to the training data.
> We will revise the title of Table 1 to clarify this distinction and prevent any potential confusion. Thank you again for your valuable suggestion.
>
> > *Q2: Some details of the training regimen are a bit unclear, e.g. **when typical backpropagation is used** and for how many training iterations.*
>
> Thank you for your careful review. We agree that some training details were too concise and may have led to confusion.
> **Backpropagation is employed both to update the learngenes, enabling them to acquire knowledge from the training classes, and to train critic networks for evaluating the generality of learngenes.**
> This process is illustrated by the arrows between Figure 3(b)(c) and (a)(e). We will revise the text and figure annotations to clarify this.
>
> In Section 4, we briefly mention the required number of training epochs, where **for medium-sized networks, only 15 epochs are needed for a 5-way classification task**. Detailed hyperparameters for training and evolving learngenes in each network are provided in Appendix Tables 6 and 7.
> We will also review and include any missing hyperparameters to further improve clarity. Thank you again for your meticulous feedback.
>
> > *Q3 It would be nice to show these results hold in the event that equation 12 in the appendix is relaxed, i.e. that **attention heads do not have matching key, query, and value matrices**.*
>
> Thank you for the insightful comment. Learngenes are constructed as parameter subsets that encode core, transferable knowledge within neural networks.
> **Equation (12) imposes a structural prior to ensure continuity of feature propagation—i.e., maintaining valid neural circuits across layers.**
> This constraint is crucial for isolating core representations from non-learngene components, thereby preserving their task-agnostic nature and compositional integrity.
>
> In response to your suggestion, we explored a relaxed setting without the structural constraint.
> **We hypothesize that tournament-based evolutionary selection may implicitly favor learngenes that maintain inter-layer connectivity, thus enforcing the constraint in a self-organizing manner.**
> As shown in the table below, even without an explicit prior, the evolution process consistently produces high-quality learngenes, underscoring the robustness and implicit regularization effects of our framework.
>
> ||Flower|CUB|Cars|Food|*Aver.*|
> |-|-|-|-|-|-|
> |He-Init|40.90|18.29|8.56|55.49|*30.81*|
> |Auto-LG|61.80|43.44|46.11|68.92|*55.07*|
> |ECO-w/o constrait|74.23*|51.13*|51.20*|69.05*|*61.40**|
> |ECO|**74.91**|**51.55**|**51.95**|**69.41**|***61.88***|
>
> > *Q4: The following papers should probably be cited: Transfer Learning Layer Selection Using Genetic Algorithm*
>
> Thank you for pointing this out. This is an important work on using evolutionary algorithms to select transferable layers in transfer learning, and we will include it in the Related Work section.
>
> > *Q5: When is **supervised learning performed** in the training loop?*
>
> Thank you for your question. As stated in our response to Q2, supervised learning is employed to train learngenes on the training classes and evaluate them on the validation classes. This process is depicted by the arrows between Figure 3(b)(c) and (a)(e), and we will clarify this further in the text and figure.
>
> > *Q6: What was the rationale behind the decision to **constrain the attention matrices** as in eq. 12?*
>
> Thank you for your insightful question. We apologize for any confusion caused by our wording.
> Our intention was not to enforce identical parameters across all attention heads but to **ensure that the corresponding learngene parameters in the query ($Q$), key ($K$), and value ($V$) matrices share the same position**.
> In other words, Eq. (12) should be more accurately written as $\mathcal{W}^q = \mathcal{W}^k = \mathcal{W}^v$, where $\mathcal{W}^q$ denotes **the set of indices** representing the learngene neurons in $Q$.
>
> **This ensures that during the continuous mapping of core features, the learngene parameters remain separate from the non-learngene components, thus preserving the integrity of the learned core knowledge.** Specifically, in the attention computation  (i.e., $A_i = \text{softmax}\left(\frac{Q_i K_i^\top}{\sqrt{d}}\right) V_i$), this constraint ensures that **the learngene parameters in $Q$, $K$, and $V$ are not multiplied with the non-learngene parameters**, preserving their distinct functionality in the attention mechanism.
> We hope this clarification addresses your concerns, and we will provide further details in the revised manuscript.
>
> Additionally, as suggested, we relaxed this constraint (as detailed in our response to Q3) and found that our evolutionary framework can implicitly preserve it through tournament-based selection. **Even without explicitly enforcing positional alignment across $Q$, $K$, and $V$, the evolution process naturally retains learngenes that satisfy this structure.** This highlights the robustness and implicit regularization capabilities of our framework.
>
> > *Q7: Why are learngenes **structurally mutated** before initializing the critic?*
>
> Thank you for your constructive question. The critic network is used to evaluate the generality of the knowledge encoded in learngenes.
> In GTL, the learngene parameters are updated through two complementary mechanisms:
> - Continuous gradient-based updating: This mechanism is applied to training classes and **is more efficient than traditional search-based approaches**. However, it may limit diversity and **risk getting trapped in local minima**, hindering exploration of a broader range of knowledge representations.
>
> - Structural mutation: To address these limitations, we introduce structural mutations, such as adding or removing kernels. **This discrete mutation introduces diversity into the learning process, allowing the learngenes to escape local minima** and explore a wider set of potentially more generalizable representations.
>
> The combination of these two mechanisms—gradient-based updating for efficiency and structural mutation for diversity—facilitates a more robust and dynamic evolution of the core knowledge encoded in learngenes, promoting improved generalization across tasks.
>
> > *Q8: Do the authors know **what is being learned at the circuit level?** A follow-up paper that determines exactly what ECO extracts would be of great interest.*
>
> Thank you for the insightful question. In response, we conducted a detailed analysis of the internal properties of learngene kernels.
> While *the NeurIPS rebuttal policy limits the inclusion of figures*, we provide supporting quantitative results here and will include visualizations in revision.
>
> - **Compact Representation**: We analyzed the weight magnitude distribution of learngene kernels and found that 68.5% of their values fall below 0.01, compared to only 39.7% in standard pre-trained models. This suggests that **learngenes are more compact, likely encoding essential transferable knowledge with reduced redundancy**.
>
>     ||Percentage of Weights < 0.01
>     |-|-
>     |Pre-trained Model|39.7
>     |Learngene|68.5
>
> - **Low-level Features**: We evaluated learngene activations across multiple tasks and observed higher cross-task similarity than in pre-trained models, suggesting that **learngenes consistently respond to shared visual features**.
> These features exhibit clear low-level characteristics, as reflected by higher similarity in early layers (more visibly demonstrated in the visualizations).
>
>     ||Cross-task Similarity
>     |-|-
>     |Pre-trained Model|24.0
>     |Learngene|45.3
>
> - **Near-Orthogonality**: We computed pairwise similarity between kernel weights and found exhibit significantly lower average off-diagonal similarity, indicating near-orthogonality. This suggests that **each kernel encodes distinct, non-redundant information, enhancing feature diversity and minimizing overlap**.
>
>     ||Off-diagonal Similarity
>     |-|-
>     |Pre-trained Model|8.69e-2
>     |Learngene|1.40e-2

---

### Official Review · Reviewer_fHjf · 2025-07-02

**Clarity:** 4
**Significance:** 4
**Originality:** 4
**Rating:** 5
**Confidence:** 3

**Summary:**

This paper proposes an evolutionary method, ECO, that operates by selection of subnetworks called learngenes. For CNNs a learngene is defined by a subset of filters at each layer and the corresponding learned kernels between them. Evolution is both Darwinian in that successful learngenes are passed on with mutations, and Lamarckian in that learned adaptations during training are inherited.

**Questions:**

When mutation adds a filter to a learngene how are the weights of that filter within the learngene initialized? Does it inherit the learned weights of the host network?

Eq 5 implies larger learngenes are more likely to grow, making the dynamics unstable. Is this as intended?

What about a steady downward pressure on learngene size, meaning the probability of mutation by removal is always greater than the probability of mutation of addition? This might act as regularization to encourage more compact learngenes.

Are the critic networks trained on the full validation set, i.e. including all n_v classes?

Are the weights within the learngene adaptable during training of the critic network?

Can filters that are part of the learngene take input from prior-layer filters that are not part of the learngene, or are those kernels fixed to zero? In the former case are those learned weights simply dropped during inheritance?

Is it possible for a learngene’s filter set to be empty at some layer? Maybe this would be useful since it could allow a learngene to be agnostic about how information is processed in the first few and last few layers, which are probably more task-specific than middle layers.

I think it would be useful to compare ECO to a single model with the same total training time, or to a conventional ensemble. For example VGG11 and ResNet12 received 250 x 20 x 15 = 75,000 total epochs of training (plus critic training on the validation set). With such a large compute budget it's important to verify that simpler methods would not perform as well.

Can you give more support for the claim that “total computational cost [is] comparable to training a typical medium-scale model”? A bit of research suggests training a single ResNet12 on ImageNet would take on the order of 100 epochs.

I’m not familiar with most of the baseline methods so I’m unclear on whether they also require training hundreds of separate networks. Do the comparisons of FLOPs and parameter counts account for these factors?

Have you experimented with multiple learngenes per network? This seems feasible and could unlock the parallel and conjunctive search that makes genetic algorithms so powerful.

**Ethical Concerns:**

["NO or VERY MINOR ethics concerns only"]

**Final Justification:**

Novel evolutionary training method, strong experimental results on multiple training and transfer datasets and diverse architectures, and compute cost comparable to baselines.

**Limitations:**

yes

**Quality:**

4

**Strengths And Weaknesses:**

Strengths:

Novel and clever evolutionary training method.

Extensive testing on multiple training and transfer datasets and diverse architectures shows significant gains over baselines in both improved performance and fewer transfer parameters.

Strong transfer across downstream tasks and changes to network architecture.

(Note: I don't follow the evolutionary computation literature closely so am not confident about the novelty or the choice of baselines. My evaluation is primarily based on the conceptual and mathematical coherence of the proposal and the strong performance relative to the reported baselines.)

Weaknesses:

High compute cost. It's not clear from the experiment results that the large number of parallel models and the large number of generations are paying off relative to non-evolutionary methods.

---

> ### Author Rebuttal · Authors · 2025-07-30
>
> Dear Reviewer fHjf,
>
> Thank you for your kind and thorough review, and for recognizing the novelty and contributions of our work.
> Your thoughtful evaluation and constructive feedback have been invaluable in helping us refine and improve our approach. Please find our detailed responses below.
>
> ---
>
> > *Q1: When mutation adds a filter to a learngene **how are the weights of that filter within the learngene initialized**? Does it inherit the learned weights of the host network?*
>
> Thank you for the insightful question. In the evolutionary process, **structural mutation is employed to introduce discrete changes into the learngene**, fostering diversity and preventing over-reliance on gradient updates, which could lead to local optima.
>
> Specifically, when a new filter is added via mutation, the goal is to explore transferable knowledge that may not be accessible through gradient descent alone.
> As you correctly noted, **the newly added filter is initialized directly from the corresponding position in the full network to ensure its relevance and facilitate smooth integration into the learngene.**
>
> > *Q2-1: Eq (5) implies **larger learngenes are more likely to grow**, making the dynamics unstable. Is this as intended?*
>
> > *Q2-2: What about a steady downward pressure on learngene size, meaning the probability of mutation by removal is always greater than the probability of mutation of addition?*
>
> Thank you for pointing this out, and we sincerely apologize for the confusion. **This was indeed a typo—we mistakenly wrote $p^+$ instead of $p^-$.**
> We understand that this could have caused confusion regarding the intended steady downward pressure on learngene size and the relative probabilities of mutation through removal versus addition.
>
> The correct equation should be:
> $p^{\scriptscriptstyle -}_l = \alpha \cdot \frac{|\mathcal{F}_l|}{N_F^{(l)} - |\mathcal{F}_l|}$
> This design aims to **balance the probabilities of adding and removing kernels, avoiding uncontrolled growth of learngenes and promoting a focus on discovering the most transferable core knowledge.**
> By maintaining this balance, learngenes can efficiently explore and adapt to useful representations within a controlled parameter budget, thereby optimizing both transferability and efficiency.
> We will correct this in revision and provide further clarification.
>
> > *Q3: Are the critic networks trained on the full validation set, i.e. including all $n_v$ classes?*
>
> Thank you for your insightful question. As you rightly pointed out, the critic network evaluates learngenes using the full set of validation classes, rather than sampling subsets of classes as in the training phase.
>
> **During evaluation, utilizing the full set of validation classes ensures consistency and fairness by eliminating task difficulty variations**, which could introduce noise or bias and potentially cause instability in the evolution process.
>
> > *Q4: Are the **weights within the learngene adaptable** during training of the critic network?*
>
> We are grateful for your valuable feedback. **During evaluation on validation classes, the weights within the learngene are updated together with the non-learngene components of the critic network**.
> In practice, it is also technically challenging to freeze specific kernels while updating others within the same filter or layer.
>
> This procedure does not compromise the validity of the evaluation. Our objective is not to measure the static performance of a fixed learngene, but to **evaluate how effectively its core knowledge supports rapid adaptation**. An optimal learngene should allow its core knowledge to adapt quickly to new tasks, thereby enabling faster convergence on unseen classes.
>
> Moreover, this process also does not affect the evolution of learngenes, as **all updates to learngenes within the critic networks are discarded after each evaluation**. Therefore, no knowledge from the validation classes is incorporated into the learngenes during evolution.
>
> > *Q5: Can filters that are part of the learngene **take input from prior-layer filters that are not part of the learngene**, or are those kernels fixed to zero? In the former case are those learned weights simply dropped during inheritance?*
>
> Thank you for pointing this out.
> During inheritance, **only the parameters corresponding to learngenes are inherited by the target network, while the non-learngene parts are randomly initialized to provide flexibility during training.**
> Specifically, to preserve core knowledge, **we initialize the non-learngene kernels in learngene filters to zero.**
> This ensures that the feature map channels corresponding to learngene filters retain only the core knowledge, preventing interference from features extracted by non-learngene filters during the early stages of training.
> After inheritance, all parameters are updated normally during training.
> Further details of learngene inheritance are provided in Appendix Figure 9.
>
> > *Q6: Is it possible for **a learngene’s filter set to be empty at some layer?***
>
> We sincerely appreciate your valuable suggestion. We agree that making learngenes agnostic to task-specific lower and higher layers is an interesting and valuable perspective. While our algorithm was not explicitly designed with this goal, it naturally allows this behavior to emerge. **Through successive kernel-removal mutations, the number of kernels in a layer can indeed be reduced to zero.**
>
> Upon examining the evolved learngenes, we observed that some indeed follow this pattern. Specifically, at the end of the evolution process, **learngenes allocate more kernels to mid-level layers, with fewer in the early and late layers.**
> We appreciate your observation and will include this in the revised manuscript.
>
> > *Q7: Can you **give more support** for the claim that “total computational cost is comparable to training a typical medium-scale model”?*
>
> Thank you for pointing out this important consideration. We provide a concrete comparison to support our claim.
> As you mentioned, training a ResNet-12 on ImageNet typically requires around 100 epochs. Given that ImageNet contains 1,000 classes, this results in approximately 100,000 class-level updates (1,000 classes × 100 epochs).
>
> In our evolutionary setup of ResNet12, each generation contains 20 networks, **each trained on a 5-way classification task** for 15 epochs. This results in 1,500 class-level updates per generation (20 network × 15 epoch × 5 classes).
> Over 250 generations, the total becomes 375,000 class-level updates (250 generation × 1,500).
>
> However, due to the smaller network size and **the parallel nature of population-based training**, this cost is distributed across devices. For example, on a single 24GB RTX 4090, we can train 4 networks in parallel, effectively reducing the cost to 93,750 class-level updates (375,000 / 4)—comparable to the 100,000 updates in standard training.
>
> ||#Class|#Epoch|#Model|#Generation|Total Class-Level Updates| Parallelism|Final Class-Level Updates|
> |-|-|-|-|-|-|-|-|
> |Standard Training|1000|100|1|–|100,000|×1|100,000|
> |Ours|5|15|20|250|375,000|×4|93,750|
>
> > *Q8: I think it would be useful to **compare ECO to a single model with the same total training time**.*
>
> Thank you for your valuable suggestion. As discussed in Q7, the overall training cost of our method is comparable to that of training a medium-scale model, due to the few-way setting used for training individual networks and the parallel nature of population-based evolution.
>
> In fact, once a model has converged under standard training, extending the training time yields diminishing returns, particularly for small and medium-sized architectures.
> To demonstrate this, we trained a ResNet-12 for 5× longer (~500 epochs), and the performance gain was marginal compared to the results presented in our paper. This further highlights the advantage of learngenes in efficiently transferring core knowledge across tasks.
>
> ||VGG11|ResNet12|
> |-|-|-|
> |1× Train|75.3|78.2|
> |5× Train|75.9|78.7|
> |ECO|**78.5**|**80.2**|
>
> > *Q9: Whether the baseline models require training hundreds of separate networks. Do the **comparisons of FLOPs and parameter counts account for these factors**?*
>
> Thank you for your constructive suggestion. Our method efficiently transfers core knowledge across tasks and model scales. We compare it to baselines focused on efficient initialization using pre-trained parameters, with detailed descriptions provided in Appendix C.2.
>
> Most baselines incur additional training costs, except for heuristic approaches like Heur-LG. Generally, **higher training costs correlate with better transfer performance**. To assess transfer efficiency, we report the number of transferred parameters in Tables 1 and 2.
> **FLOPs are comparable across methods, as they share the same backbone architecture**, with the notable exception of knowledge distillation, which introduces a teacher model and results in higher computational costs, as clearly indicated in Table 2.
>
> > *Q10: Have you experimented with **multiple learngenes per network***?
>
> Thanks for your constructive suggestion. **Combining multiple learngenes is feasible, especially in CNNs, where kernels naturally function as modular units.** In our original manuscript, we focused on evolving a single learngene per network, without exploring multi-learngene combinations. Following your suggestion, we conducted preliminary experiments by combining learngenes from the final generation. The results showed modest performance gains compared to using a single learngene.
>
> ||VGG11|ResNet12|
> |-|-|-|
> |Single Learngene|78.5|80.2|
> |Multiple Learngenes|79.0|80.6|
>
> However, since our initial evolution strategy does not explicitly encourage combining multiple learngenes, redundant core knowledge may be present across different learngenes.
> To address this, **we plan to extend the GTL framework to support multi-learngene combinations, similar to genetic "crossover" in biological systems.**

---

> > ### Comment · Reviewer_fHjf · 2025-08-04
> >
> > Thank you for the thorough replies. They address my concerns well. The other reviewers raise interesting points but nothing fatal and the authors offer strong replies.
> >
> > The other reviews and rebuttals make me more confident about the novelty of the approach (which I wasn't well positioned to judge). In particular ECO appears to be unique in selecting for transferable knowledge (since critics are given different tasks from learners) rather than for performance on a fixed task, as well as its combination of Darwinian and Lamarckian mechanisms.
> >
> > I have some remaining questions about compute cost, which I think all the reviewers agree is a concern. In your rebuttal you refer to class-level updates, but it's not clear why that is the right metric. Doesn't computing and updating the model with 5 output logits cost nearly as much as with 100, since the intermediate layer computations are the same? Could you provide wall-clock comparisons instead?

---

> > > ### Author Response · Authors · 2025-08-05
> > >
> > > Dear Reviewer fHjf,
> > >
> > > Thank you for your thoughtful review and for the positive recognition of our work’s novelty. We greatly appreciate the time you have taken to thoroughly read and engage with our rebuttal, as well as your continued interest in discussing the computational cost concerns.
> > >
> > > We are happy to provide further clarification on the computational cost.
> > > First, it is important to confirm that **for the same network architecture (e.g., ResNet-12), the time taken for a forward-backward pass within a batch remains consistent** whether we are training on ImageNet or performing few-way training in our evolutionary process, as the number of FLOPs is identical.
> > > However, the total training time per epoch differs significantly between these scenarios due to **the size of the training datasets**.
> > >
> > > For example, in ImageNet, there are 1,000 classes, with roughly 1280 images per class. As such, one full pass through the dataset (i.e., one epoch) covers 1,000 x 1280 = 1,280,000 images. With a batch size of 256, this results in 5,000 gradient updates (1,280,000 / 256).
> > >
> > > In contrast, during evolution, each model is trained on a 5-way task (i.e., 5-class classification), which covers 5 x 1280 = 6,400 images per epoch. With the same batch size of 256, this results in only 25 gradient updates per epoch (6,400 / 256).
> > > **Since the time for each gradient update remains constant, the number of updates per epoch is a major factor in the difference in training times across these two settings**. Therefore, the class-level updates metric serves as a reasonable measure of the relative computational cost, as it captures the amount of data being processed per epoch.
> > >
> > > We have re-evaluated the time cost in terms of gradient updates, as shown in the table below. We assume that the time for a single forward-backward pass is denoted as $t$, which remains consistent due to the identical architecture. In the case of ImageNet training, the total training time (i.e., 100 epoch) is approximately $500,000 \times t$ (100 $\times$ 5,000 $\times t$).
> > >
> > > In contrast, during evolution, there are 250 generations, each consisting of 20 networks, and each network undergoes 15 epochs, resulting in a total of 75,000 epochs. Each epoch involves 25 gradient updates, which totals approximately $1,875,000 \times t$ ($75,000 \times 25 \times t$) in training time.
> > >
> > > However, due to the smaller network size and the parallel nature of population-based training, this cost is distributed across multiple devices. For instance, on a single 24GB RTX 4090, we can train 4 networks in parallel, effectively reducing the overall cost to $468,750 \times t$ ($1,875,000/ 4 \times t$), which is comparable to the $500,000 \times t$ updates in standard ImageNet training.
> > >
> > > ||#Class|#Training Data|#batchsize|*#Gradient Updates Per Epoch*|#Epoch|#Model|#Generation|*#Total Epoch*|#Total Gradient Updates| Parallelism|#Total Time|
> > > |-|-|-|-|-|-|-|-|-|-|-|-|
> > > |Standard Training|1000|1280×1000|256|***5,000***|100|1|–|***100***|500,000|×1|500,000 $\times$ $t$|
> > > |Ours|5|1280×5|256|***25***|15|20|250|***75,000***|1,875,000|×4|468,750 $\times$ $t$|
> > >
> > > ---
> > >
> > > To further clarify, we have provided the following wall-clock comparisons, which is trained on a single RTX 4090:
> > >
> > > When training ResNet-12 on ImageNet, one epoch requires approximately 10 minutes, with a forward-backward pass taking around 0.12 seconds (i.e., $t \approx$ 0.12 seconds).
> > > In comparison, training one generation of models (20 models) within our evolutionary framework takes approximately 3.75 minutes.
> > >
> > > Therefore, the total wall-clock time for training 100 epochs on ImageNet is approximately 1,000 minutes, while training 250 generations of models in evolution requires approximately 937 minutes.
> > >
> > > We believe these comparisons provide clearer insight into the time efficiency of our approach. We hope this explanation effectively addresses your concerns, and we sincerely appreciate your thoughtful feedback and continued engagement with our work.
> > >
> > > Best regards,
> > >
> > > Authors

---

> > > > ### Comment · Reviewer_fHjf · 2025-08-05
> > > >
> > > > Got it! Maybe describe the comparisons in terms of # examples processed (or minibatches). That plus the wall clock results make it clear your method is not expensive compared to single model training.

---

> > > > > ### Author Response · Authors · 2025-08-05
> > > > >
> > > > > Dear Reviewer fHjf,
> > > > >
> > > > > We're pleased to hear that our explanation helped address your concerns. We will certainly revise our manuscript to include the requested comparisons in terms of the number of examples processed and minibatches.
> > > > >
> > > > > If you have any further questions or suggestions, please don't hesitate to reach out. We truly appreciate your thoughtful engagement and look forward to refining our work based on your valuable feedback.
> > > > >
> > > > > Best regards,
> > > > >
> > > > > Authors

---

### Official Review · Reviewer_aR1z · 2025-07-02

**Clarity:** 3
**Significance:** 3
**Originality:** 3
**Rating:** 5
**Confidence:** 4

**Summary:**

The authors tackle the task of transfer learning using a novel representation called learngenes. These are defined as subnetworks from optimized neural networks which are extracted to be used as population members of an evolutionary algorithm. At each iteration networks are initialized with the subnetwork defined by the learngene plus other randomly initialized units. Learngenes can be muted by expanding and reducing the units that they cover. The authors test the approach on several transfer learning benchmarks.

**Questions:**

- Are learngenes fixed to particular positions? Doesn't that limit the ability to use the alleged core knowledge in other parts of the network?
- Are there no other approaches such as NEAT or Quality-Diversity algorithm that could provide similar benefits and thus serve as useful baselines?
- What are the possible limitations of the approach? How effectively can it scale to larger networks?

**Ethical Concerns:**

["NO or VERY MINOR ethics concerns only"]

**Final Justification:**

I find the idea interesting and the authors response to my concerns sufficient to raise my score.

**Limitations:**

The authors do not mention any limitations.

**Paper Formatting Concerns:**

None.

**Quality:**

2

**Strengths And Weaknesses:**

Strengths:
- The problem is well motivated as good algorithms for transfer learning is very much an open question.
- The idea is similar to approaches such as NEAT, but manages to integrate it well with gradient descent approaches that have proven so successful.

Weaknesses:
- The representation is not general, and must in fact be defined for each possible architecture one wishes to use.
- I find some the way learngenes are described to be a bit misleading. For example the authors claim that they are optimized for knowledge transfer, not task performance, but it appears to me that it is indeed the later which they do. This is based on the fact that they are optimized and selected for their performance. Knowledge transfer is not optimized for, but just a consequence of the compressed nature of the learngene.
- Similarly, core knowledge seems to be a weird way to describe what is found by learngenes. In fact it looks more like finding the lottery ticket hypothesis network within the overall architecture.

---

> ### Author Rebuttal · Authors · 2025-07-30
>
> Dear Reviewer aR1z,
>
> We sincerely thank you for your thoughtful and encouraging review.  We appreciate your recognition of our motivation and the positive assessment of the GTL framework.
> Below, we provide detailed responses to your comments.
>
> ---
>
> > *Q1: The representation is not general, and **must in fact be defined for each possible architecture** one wishes to use.*
>
> Thank you for your insightful comment. This work explores how transferable core knowledge can be decoupled from neural networks and encapsulated into reusable learngenes for efficient transfer.
> We evaluate learngenes for both CNN and Transformer architectures, **with a unified representation that generalizes well across diverse model families.**
>
> Specifically, for CNNs, learngenes are constructed at the level of convolutional kernels—**the fundamental units in modern architectures** for capturing localized visual patterns.
> Although architectures like GhostNet and EfficientNet introduce advanced modules (e.g., Ghost modules and SE-enhanced MBConv blocks), they retain the core convolutional structure, thereby **preserving the effectiveness of kernel-level learngenes for knowledge transfer.**
>
> To validate the applicability of learngenes in modern CNNs, we extend them on GhostNet-v2 and EfficientNetV2.
> As shown in table below, learngenes remain compatible and exhibit strong transferability despite architectural innovations.
> |GhostNet|Flower|CUB|Cars|Food
> |-|-|-|-|-
> |He-Init|44.46|50.35|64.79|72.33
> |Auto-LG|60.72|56.47|70.54|73.79
> |ECO|**67.44**|**61.70**|**75.40**|**74.27**
>
> |EfficientNet|Flower|CUB|Cars|Food
> |-|-|-|-|-
> |He-Init|49.47|47.08|65.15|68.74
> |Auto-LG|65.58|59.60|74.83|74.13
> |ECO|**71.08**|**64.58**|**79.72**|**77.97**
>
> > *Q2: The authors claim that they are **optimized for knowledge transfer**, not task performance, but it appears to me that it is indeed the later which they do.*
>
> Thank you for the valuable question. The Genetic Transfer Learning (GTL) optimizes knowledge transfer by addressing a key challenge in transfer learning: **identifying what knowledge should be transferred to maximize benefit and efficiency, particularly under large domain shifts**.
>
> To this end, we focus on isolating **a compact subset of parameters that encapsulates highly transferable, task-agnostic knowledge for efficient transfer**, referred to as learngenes.
> However, in conventionally pre-trained models, such knowledge is **typically entangled throughout the parameter space**, hindering its extraction.
>
> To explicitly decouple and encapsulate transferable knowledge for efficient reuse, we propose the GTL framework, which introduces a genetic mechanism into neural networks.
> **In GTL, learngenes are continuously transferred across networks and tasks, during which the core knowledge they encode is incrementally optimized and accumulated.**
>
> This process gradually disentangles the most essential transferable components from the full network and encapsulates them within the learngene.
> Thus, **GTL optimizes the transferability of knowledge itself encoded in leanrgenes**, rather than directly maximizing task performance within a single training instance.
>
> > *Q3: Core knowledge seems to be a weird way to describe what is found by learngenes. In fact it **looks more like finding the lottery ticket hypothesis network** within the overall architecture.*
>
> We appreciate the connection to the Lottery Ticket Hypothesis (LTH), which posits that a dense network contains a sparse subnetwork that can match its performance when trained independently with the same initialization.
> **Although both LTH and Learngene involve identifying parameter subsets within networks, they differ fundamentally in motivation, mechanism, and outcome.**
>
> The key distinctions are summarized below:
>
> - **Fundamental Motivation**:
>   - **LTH**: Identifies **task-specific subnetworks** that are easier to train, but are effective only when retrained with the same initialization and data, and therefore **lack cross-task or cross-domain transferability** (see table below).
>     ||Flower|CUB|Cars|Food
>     |-|-|-|-|-
>     |LTH|54.76|57.28|70.89|72.93
>     |ECO|**77.02**|**65.12**|**82.61**|**76.13**
>
>   - **Learngene**: Encodes **task-agnostic core knowledge** optimized for cross-task and cross-domain transfer, and supports **scalable adaptation** across networks of varying sizes.
>
> - **Optimization Target**:
>
>   - **LTH**: Seeks to **reduce parameter count** while preserving performance on a fixed task, effectively optimizing for **model compression**.
>
>   - **Learngene**: Seeks to **maximize knowledge reuse** across diverse tasks with minimal parameter overhead, focusing on **transfer efficiency** rather than compression.
>
> - **Discovery Mechanism**:
>
>   - **LTH**: Discovers sparse subnetworks **after full training via pruning and rewinding**, relying on static analysis.
>
>   - **Learngene**: Constructed and refined **during training through evolutionary inheritance across diverse networks and tasks**, enabling continual accumulation of core transferable knowledge.
>
> > *Q4: Are learngenes **fixed to particular positions**? Doesn't that limit the ability to **use the alleged core knowledge in other parts of the network?***
>
> Thank you for pointing this out. Due to the translational invariance property of CNNs, the ordering of kernels within a layer is not fixed.
> **As demonstrated in Appendix Figure 10, reordering kernels does not affect functionality as long as the corresponding input channels in subsequent layers are adjusted accordingly.** Therefore, learngenes are not tied to specific kernel positions and can be flexibly adapted to networks of varying depth or width.
>
> We then explain *how non-learngene components benefit from core knowledge*. In CNNs, when the $i$-th filter in layer $l$ includes learngene kernels, the resulting $i$-th channel in the feature map encodes core representations. **This channel is subsequently processed by all filters in the next layer—regardless of whether they belong to a learngene—allowing core knowledge to propagate throughout the network.**
>
> > *Q5: Are there no other approaches such as **NEAT or Quality-Diversity algorithm** that could provide similar benefits?*
>
> We appreciate your valuable suggestion. As noted in our response to Q2, **our method is not a traditional search-based evolutionary algorithm (e.g., NEAT)**, but departs from conventional evolutionary algorithms (EAs) in several fundamental ways:
>
> - **Optimization Objective:**
>
>   - **EA**: Aims to **search for task-specific optimal solutions** by evolving populations of candidate solutions for a given task.
>
>   - **GTL**: Aims to **decouple task-agnostic core knowledge** to enable efficient adaptation to novel tasks across diverse domains.
>
> - **Gene Form and Function:**
>
>   - **EA**: "Genes" refers to **encodings of a candidate solution**—typically real-valued vectors (e.g., in Evolution Strategies) or binary strings (e.g., in Genetic Algorithms).
>
>   - **GTL**: Learngenes are **network fragments that encapsulate transferable knowledge**.
>   Rather than encoding solutions, they serve as knowledge carriers that facilitate rapid adaptation in new networks.
>
> - **Evolutionary Mode:**
>
>   - **EA**: Follows Darwinian principles, evolving populations through mutation and crossover to **explore the solution space**.
>
>   - **GTL**: Adopts a Lamarckian approach, where knowledge acquired through gradient-based learning is retained and passed on via learngenes, enabling **cumulative transfer across tasks**.
>
> Building on this, learngene fundamentally differs from traditional search-based methods such as NEAT or Quality-Diversity (QD). While these methods can search over network structures or parameters, they primarily target task-specific optimization and are often computationally intensive, especially for deep networks.
>
> **Instead, we include a more relevant baseline tailored to transfer learning: P-Transfer, which employs evolutionary strategies to identify the most suitable layers for fine-tuning.**
> As shown in the table below, despite transferring all parameters, P-Transfer performs poorly under large domain shifts, supporting our central claim that full-model transfer is suboptimal under substantial distributional differences.
>
> |VGG11|Param.|Flower|CUB|Cars|Food
> |-|-|-|-|-|-
> |He-Init|0M|34.22|49.81|60.94|78.67
> |P-Transfer|28.11M|**72.32**|52.21|72.37|61.54
> |ECO|1.94M|64.42|**60.20**|**78.12**|**79.47**
>
> |ResNet12|Param.|Flower|CUB|Cars|Food
> |-|-|-|-|-|-
> |He-Init|0M|50.74|46.89|61.27|81.10
> |P-Transfer|12.43M|81.01|63.12|83.27|79.56
> |ECO|2.47M|**81.48**|**64.14**|**84.28**|**82.41**
>
> > *Q6: What are the possible limitations of the approach? How effectively can it **scale to larger networks?***
>
> Thanks for your question.
> Our main contribution is showing that transferring task-agnostic core knowledge, rather than the full model, yields better transferability and efficiency.
> We also acknowledge the following limitations, as noted in Appendix E.
>
> Due to computational constraints, our experiments focus on small- to medium-scale networks, and we have not yet extended GTL to large architectures such as GPT-2.
> Nevertheless, as discussed in the paper, **the overall cost of the evolutionary process is comparable to standard full-model training.**
> Furthermore, **GTL supports scalability via population-level parallelism and the few-shot nature of individual tasks, enabling application to larger models without significant additional overhead.**
>
> That said, we acknowledge that increasing model depth and width—particularly in CNNs—results in exponential growth in kernel count, introducing greater structural uncertainty during learngene optimization.
> **To address this, future work may explore coarser learngene granularity, such as using entire filters instead of individual kernels, thereby reducing search space complexity and improving scalability.**
>
> We will elaborate on potential extensions to large-scale networks in the revised manuscript.

---

> > ### Comment · Reviewer_aR1z · 2025-08-05
> >
> > I thank the authors for the detailed response, especially the explanation of how the learngenes work and how they can transfer knowledge. I would encourage the authors to incorporate some of their own response into the final text. Keeping track of how the largeness are supposed to work is not completely intuitive.
> >
> > In any case, I am satisfied with this reply and will raise my score.

---

> > > ### Author Response · Authors · 2025-08-05
> > >
> > > Dear Reviewer aR1z,
> > >
> > > Thank you for your generous increase in rating and for your thoughtful feedback. We truly appreciate the time and effort you have invested in reviewing our paper.
> > >
> > > As per your suggestion, we will incorporate the relevant portions of our response into the final manuscript to further clarify the workings of the learngenes and their knowledge transfer mechanisms.
> > >
> > > Once again, we sincerely thank you for your valuable time and constructive input.
> > >
> > > Best regards,
> > >
> > > Authors

---

### Official Review · Reviewer_evzs · 2025-07-03

**Clarity:** 3
**Significance:** 2
**Originality:** 2
**Rating:** 4
**Confidence:** 4

**Summary:**

This paper introduces ECO (Evolving Core Knowledge), a novel framework designed to improve the efficiency and flexibility of knowledge transfer in neural networks. The work draws inspiration from biological evolution, proposing a method to distill essential, reusable knowledge into modular components called "learngenes".

The authors argue that existing transfer learning methods, which often rely on reusing entire sets of parameters, are inefficient and lack adaptability across different model architectures and tasks. To address these limitations, ECO introduces a paradigm called Genetic Transfer Learning (GTL). GTL simulates an evolutionary process where a population of neural networks is trained on a variety of tasks. High-performing networks are selected, and their core knowledge, encapsulated in "learngenes", is passed on to the next generation. These learngenes are defined as interconnected neural circuits, such as sets of kernels in CNNs.

Through iterative cycles of inheritance, mutation, and selection, GTL is designed to refine these learngenes, ensuring they contain compact, task-agnostic, and transferable knowledge. This approach allows for the initialization of diverse neural network architectures, of varying sizes, with this evolved core knowledge, significantly reducing computational and memory costs compared to traditional pre-training and fine-tuning.

The paper presents extensive experiments across a range of datasets, including CIFAR-FS, miniImageNet, and ImageNet, using various architectures including VGG, ResNet, MobileNetV3, and ViT. The results demonstrate that ECO consistently outperforms existing methods in terms of both accuracy and resource efficiency. Notably, it can achieve comparable or even superior performance to full fine-tuning while drastically reducing the number of transferred parameters.

**Questions:**

1. Considering the significant upfront investment for the evolutionary process, have you analyzed the break-even point where the benefits of reusing the efficient learngenes outweigh the initial training cost? For example, how many downstream models must be initialized with a learngene to make it more cost-effective than standard pre-training approaches?

2. The CAM visualizations effectively show that learngenes help models focus on semantically meaningful regions. Beyond these visualizations of their effects, have you conducted any preliminary analysis into the intrinsic properties of the learngenes themselves? For instance, do the evolved kernels represent recognizable low-level features like edge detectors or texture filters in early layers?

3. Please also address the weakness comments.

**Ethical Concerns:**

["NO or VERY MINOR ethics concerns only"]

**Final Justification:**

Authors' responses and additional results (especially case studies and ablations) provided have address my concerns.

**Limitations:**

Please see the weakness points above.

**Quality:**

2

**Strengths And Weaknesses:**

S1. The analogy to biological genetics is not merely a superficial inspiration but is thoughtfully integrated into the technical framework of GTL. The concepts of "learngenes", mutation, inheritance, and selection are well-defined and operationalized for neural networks. This provides an elegant solution to the problem of identifying and isolating transferable knowledge.

S2. The paper provides a robust and comprehensive evaluation of ECO. The consistent outperformance across a variety of datasets, architectures, and tasks, including few-shot learning and downstream applications, lends strong support to the authors' claims. The demonstrated efficiency in terms of reduced parameter transfer and computational overhead is particularly noteworthy.

W1. While the resulting learngenes are efficient to use, the evolutionary process itself appears computationally intensive. The paper notes that the total cost is "comparable to training a typical medium-scale model," but evolving learngenes for hundreds of generations with a population of networks is a significant upfront investment. Further analysis of the trade-offs between the cost of evolution and the benefits of the resulting learngenes would be beneficial.

W2. The GTL framework introduces a new set of hyperparameters related to the evolutionary process, such as population size, mutation rates, and selection mechanisms. The sensitivity of the final learngene quality to these parameters could be a crucial factor in the practical application of this method. A more detailed ablation study on these evolutionary hyperparameters would strengthen the paper.

W3. While the paper visualizes the effects of learngenes on model attention, the intrinsic meaning of the knowledge encoded within a learngene is still largely a black box. Future work could delve into methods for interpreting what specific features or computational primitives these evolved neural circuits represent.

W4. I guess similar mechanism as been studied in "Delta: Deep Learning Transfer using Feature Map with Attention for Convolutional Networks", ICLR 2019. Authors in this paper also proposed a method to select CNN kernels with discriminative powers and conduct knowledge distillation from the visual features generated by these kernels for transfer learning.

W5. While the paper provides a comprehensive evaluation across several foundational models, the selection of architectures could be considered somewhat dated, which may temper the broader claims of scalability and generalizability. The study primarily relies on architectures such as VGG11, ResNet, and MobileNetV3. Although these models are undoubtedly influential and serve as important benchmarks, the field of neural architecture design has evolved significantly. Newer architectures often incorporate more advanced features such as sophisticated attention mechanisms, dynamic convolutions, and more efficient scaling strategies that are not present in these older designs.

---

> ### Author Rebuttal · Authors · 2025-07-30
>
> Dear Reviewer evzs,
>
> Thank you for your careful review of our manuscript. We sincerely appreciate your recognition of the biological inspiration behind our method, as well as your acknowledgment of the comprehensiveness of our evaluation.
> Below, we provide detailed responses to your comments.
>
> ---
>
> > *Q1: While the resulting learngenes are efficient to use, the **evolutionary process itself appears computationally intensive**.*
>
> Thank you for the valuable question. To clarify, when we state that "the total cost is comparable to training a typical medium-scale model," we mean that **the overall computational cost of the learngene evolution** is similar to pre-training a model of the same size from scratch.
>
> While evolutionary methods are often computationally intensive and rarely used for network optimization, our approach is designed to minimize such costs.
> - Instead of training on the full dataset, **each network solves a lightweight few-shot task by sampling $n$-way classification subsets (typically $n=5$)**, improving efficiency and facilitating the discovery of shared core knowledge.
>
> - Instead of **performing traditional parameter search**, our evolution establishes a genetic mechanism for knowledge transfer. Learngenes are inherited and refined across networks through gradient-based updates, enabling efficient accumulation of transferable knowledge.
>
> To clarify the computational cost of learngene evolution, we provide a simple comparison.
> For instance, conventional training of ResNet-12 on ImageNet typically involves 100 epochs.
> Given that ImageNet contains 1,000 classes, this results in approximately 100,000 class-level updates (1,000 classes × 100 epochs).
>
> In our evolutionary setup of ResNet12, each generation contains 20 networks, **each trained on a 5-way classification task** for 15 epochs. This results in 1,500 class-level updates per generation (20 network × 15 epoch × 5 classes).
> Over 250 generations, the total becomes 375,000 class-level updates (250 generation × 1,500).
>
> However, due to the smaller network size and **the parallel nature of population-based training**, this cost is distributed across devices. For example, on a single 24GB RTX 4090, we can train 4 networks in parallel, effectively reducing the cost to 93,750 class-level updates (375,000 / 4)—comparable to the 100,000 updates in standard training.
>
> ||#Class|#Epoch|#Model|#Generation|Total Class-Level Updates| Parallelism|Final Class-Level Updates|
> |-|-|-|-|-|-|-|-|
> |Standard Training|1000|100|1|–|100,000|×1|100,000|
> |Ours|5|15|20|250|375,000|×4|93,750|
>
> > *Q2: The GTL framework introduces a new set of hyperparameters related to the evolutionary process, such as population size. The **sensitivity of the final learngene quality to these parameters** could be a crucial factor in the practical application of this method.*
>
> Thank you for your valuable suggestions. As noted in our response to Q1, GTL departs from traditional search-based evolutionary algorithms by focusing on knowledge transfer and refinement rather than explicit evolutionary search.
> **Consequently, GTL is relatively insensitive to hyperparameter choices.**
>
> In response to your suggestion, we conducted additional experiments on population size, mutation rate, and selection strategy.
> **The results confirm that learngene quality remains robust across these variations, highlighting the stability of our evolution process centered on knowledge transfer.**
>
> |Population Size|2|10|*20**|40
> |-|-|-|-|-
> |VGG11 Acc.|75.9|76.8|77.0|77.1
> |ResNet12 Acc.|80.7|81.5|81.6|81.6
>
> |Mutation Rates|0.1|*0.2**|0.3|0.5
> |-|-|-|-|-
> |VGG11 Acc.|76.4|77.0|76.9|76.5
> |ResNet12 Acc.|80.9|81.6|81.4|80.2
>
> |Selection Mechanisms|Roulette Wheel|Elitism|*Tournament**
> |-|-|-|-
> |VGG11 Acc.|77.1|76.8|77.0
> |ResNet12 Acc.|81.5|81.4|81.6
>
> > *Q3: While the paper visualizes the effects of learngenes on model attention, the intrinsic meaning of the **knowledge encoded within a learngene** is still largely a black box.*
>
> Thank you for the insightful question. In response, we conducted a detailed analysis of the internal properties of learngene kernels.
> While *the NeurIPS rebuttal policy limits the inclusion of figures*, we provide supporting quantitative results here and will include visualizations in revision.
>
> - **Compact Representation**: We analyzed the weight magnitude distribution of learngene kernels and found that 68.5% of their values fall below 0.01, compared to only 39.7% in standard pre-trained models. This suggests that **learngenes are more compact, likely encoding essential transferable knowledge with reduced redundancy**.
>
>     ||Percentage of Weights < 0.01
>     |-|-
>     |Pre-trained Model|39.7
>     |Learngene|68.5
>
> - **Low-level Features**: We evaluated learngene activations across multiple tasks and observed higher cross-task similarity than in pre-trained models, suggesting that **learngenes consistently respond to shared visual features**.
> These features exhibit clear low-level characteristics, as reflected by higher similarity in early layers (more visibly demonstrated in the visualizations).
>
>     ||Cross-task Similarity
>     |-|-
>     |Pre-trained Model|24.0
>     |Learngene|45.3
>
> - **Near-Orthogonality**: We computed pairwise similarity between kernel weights and found exhibit significantly lower average off-diagonal similarity, indicating near-orthogonality. This suggests that **each kernel encodes distinct, non-redundant information, enhancing feature diversity and minimizing overlap**.
>
>     ||Off-diagonal Similarity
>     |-|-
>     |Pre-trained Model|8.69e-2
>     |Learngene|1.40e-2
>
> > *Q4: I guess similar mechanism as been studied in "Delta: Deep Learning Transfer using Feature Map with Attention for Convolutional Networks".*
>
> Thank you for your insightful question. **DELTA focuses on identifying kernels most relevant to a specific target dataset, which contrasts with our task-agnostic, generalizable approach.**
>
> As a distillation-based method, DELTA performs **task-specific transfer from one source to one target domain**, retaining only features useful for that task.
> However, this limits generality: **the transferred knowledge is tailored to a single task and cannot be reused across tasks**.
> As a result, DELTA must be repeated $n×$ for $n$ downstream datasets, reducing scalability.
>
> In contrast, learngenes support more general and efficient transfer by **extracting task-agnostic core knowledge applicable across diverse tasks**.
> Even with multiple downstream tasks, **they can be directly used for initialization without incurring additional overhead** (e.g., no repeated evolution).
>
> > *Q5: While the paper provides a comprehensive evaluation across several foundational models, the **selection of architectures could be considered somewhat dated**, which may temper the broader claims of scalability and generalizability.*
>
> Thank you for pointing this out. While models like VGG and ResNet may not reflect recent architectural advances, **this does not affect the scalability or generalizability of learngenes.**
>
> In CNNs, learngenes are constructed at the level of convolutional kernels, which **remain fundamental computational units in modern architectures**.
> Although recent models incorporate advanced mechanisms—such as attention, dynamic convolutions—convolutional kernels continue to play a central role in capturing localized visual patterns (e.g., edges, textures).
> **This structural consistency ensures that learngenes retain their validity for knowledge transfer, even as architectural designs evolve.**
>
> To validate this, we evaluate learngenes on GhostNet-v2 and EfficientNetV2, which incorporate DFC attention and compound scaling.
> As shown in the table below, despite these enhancements, both architectures retain the core convolutional paradigm, **allowing kernel-level learngenes to remain effective with strong compatibility and transferability**.
>
> |GhostNet|Flower|CUB|Cars|Food
> |-|-|-|-|-
> |He-Init|44.46|50.35|64.79|72.33
> |Auto-LG|60.72|56.47|70.54|73.79
> |ECO|**67.44**|**61.70**|**75.40**|**74.27**
>
> |EfficientNet|Flower|CUB|Cars|Food
> |-|-|-|-|-
> |He-Init|49.47|47.08|65.15|68.74
> |Auto-LG|65.58|59.60|74.83|74.13
> |ECO|**71.08**|**64.58**|**79.72**|**77.97**
>
> > *Q6: Considering the significant upfront investment for evolutionary process, have you **analyzed the break-even point** where the benefits of reusing the efficient learngenes outweigh the initial training cost?*
>
> Thank you for your valuable comments. We understand the concern regarding the computational overhead of evolutionary methods.
> However, in GTL, **each network is trained on a few-way classification task using a small subset of training data, and the population can be trained in parallel, making the overall cost comparable to standard pre-training** (see analysis in Q1).
>
> As network size increases, population-level parallelism may be limited by GPU memory. For instance, we can parallelize 4 networks per GPU for VGG11 and ResNet12, but only 2 for larger models like ResNet50 and ViT.
> Nevertheless, the additional computational cost remains modest—much less than double the original.
> Thus, **even for large networks, initializing just two downstream networks is sufficient to reaches a break-even point, while for medium- and small-scale models, initializing a single downstream model is sufficient.**
>
> > *Q7: The CAM visualizations effectively show that learngenes help models focus on semantically meaningful regions. Have you conducted any preliminary analysis into **the intrinsic properties of the learngenes themselves**?*
>
> Thank you for your valuable suggestions. As discussed in our response to Q3, learngenes exhibit compact representations that emphasize essential and transferable knowledge.
> They consistently capture low-level features shared across tasks—particularly in early layers. In addition, their near-orthogonal structure indicates that each kernel encodes distinct, non-redundant information. Please refer to Q3 for further details.

---

### Note · Authors · 2025-08-12

We sincerely thank the Area Chair and Reviewers for their valuable feedback, which has enhanced the rigor, depth, and clarity of our work. We are encouraged that our core contributions have been broadly recognized and we believe all concerns have been comprehensively addressed.

### **Strengths Acknowledged by Reviewers**
**Clear Motivation and Novelty**: All reviewers (evzs, aR1z, fHjf, X8DF) highly praised our biologically inspired approach to efficient neural network knowledge transfer and acknowledged its novelty.

**Comprehensive Evaluation**: Reviewers evzs and X8DF recognized the robustness and thoroughness of our experimental design.

**Strong Performance**: Reviewer aR1z highlighted the efficiency of our learngene evolution, while Reviewers fHjf and X8DF affirmed its transferability and downstream task effectiveness.

### **Concerns and Responses**
The primary concerns raised by the reviewers focused on:

- **Cost of the Evolutionary Process (Reviewers evzs and fHjf)**:
We provided detailed timing and wall-clock comparisons with traditional pre-training. We are pleased to see Reviewer fHjf indicate that these concerns have been satisfactorily addressed.

- **Generality of the Updated Network Structures (Reviewers evzs and aR1z)**:
We supplemented our manuscript with results on advanced architectures such as GhostNet and EfficientNet to further clarify the structural generality of learngenes. Reviewer aR1z expressed satisfaction with these additions.

- **Knowledge Representation within Learngenes (Reviewers evzs and X8DF)**:
We presented statistical evidence demonstrating that learngenes constitute a compact representation characterized by low-level features and near-orthogonality.

---

We appreciate Reviewer fHjf’s thorough review of other Reviewers’ concerns and our responses, noting:
> "The other reviewers raise interesting points but nothing fatal and the authors offer strong replies."

We appreciate the encouraging and positive feedback from Reviewers aR1z and X8DF.

We have also clarified Reviewer X8DF’s follow-up questions regarding validation classes; although no further discussion ensued, we believe these concerns have been adequately addressed.

We understand that Reviewer evzs may have had limited time during Discussion phase; nonetheless, we sincerely appreciate the valuable feedback he provided. Since most of his concerns overlap with those of other reviewers, we remain confident that our rebuttal has sufficiently addressed them.

---

### Decision · Program_Chairs · 2025-09-17

**Decision:**

Accept (poster)

**Comment:**

This paper introduces ECO, a novel, bio-inspired framework that evolves modular neural circuits called "learngenes" to achieve efficient and robust knowledge transfer, and evaluates the method on a variety of datasets. The proposed method presents an intriguing means of combining gradient-based training with evolutionary methods to produce qualitatively different features than what might be expected via pure gradient descent. While reviewers expressed concerns about the method’s computational cost and generality, the additional evaluations provided during the rebuttal period allayed these concerns. Overall, the paper presents an interesting and creative approach to training which, while unlikely to usurp gradient-based pretraining methods in the near future, provides a refreshing change of perspective on neural network training which will benefit the NeurIPS community.